# AdaptiveK Sparse Autoencoders: Dynamic Sparsity Allocation for Interpretable LLM Representations

## Abstract

Understanding the internal representations of large language models (LLMs) remains a central challenge for interpretability research. Sparse autoencoders (SAEs) offer a promising solution by decomposing activations into interpretable features, but existing approaches rely on fixed sparsity constraints that fail to account for input complexity. We propose **AdaptiveK SAE** (Adaptive Top K Sparse Autoencoders), a novel framework that dynamically adjusts sparsity levels based on the semantic complexity of each input. Leveraging linear probes, we demonstrate that context complexity is linearly encoded in LLM representations, and we use this signal to guide feature allocation during training. Experiments across ten language models (from 70M to 14B parameters) demonstrate that this complexity-driven adaptation significantly outperforms fixed-sparsity approaches on reconstruction fidelity, explained variance, cosine similarity and interpretability metrics while eliminating the computational burden of extensive hyperparameter tuning.

## 1 Introduction

As large language models (LLMs) continue to advance, understanding their internal representations becomes increasingly crucial yet challenging. These models operate as "black boxes" with activation spaces that resist straightforward analysis (Olah et al., 2020; Ferrando et al., 2024). Individual components typically respond to multiple unrelated concepts (polysemanticity) (Olah, 2023), while the models encode more distinct features than their dimensional capacity would suggest (superposition) (Arora et al., 2018; Gurnee et al., 2023; Elhage et al., 2022). This efficient but complex encoding creates a significant interpretability barrier as traditional approaches cannot untangle the overlapping information patterns. Sparse autoencoders (SAEs) (Bricken et al., 2023; Cunningham et al., 2023) address this challenge by decomposing model activations into sparse combinations of interpretable features, revealing the underlying structure of the model's representations.

Recent research has rapidly expanded the capabilities of sparse autoencoders, scaling them to extract millions of interpretable features from frontier models (Templeton et al., 2024) while introducing numerous architectural innovations (Rajamanoharan et al., 2024a; Gao et al., 2024; Bussmann et al., 2024a; Rajamanoharan et al., 2024b; Taggart, 2024; Mudide et al., 2024; Bussmann et al., 2024b; Karvonen et al., 2024; Marks et al., 2024a; Braun et al., 2024). However, despite these advancements, current SAE architectures rely on uniform sparsity constraints regardless of input complexity. Whether through activation-limiting approaches like TopK (Gao et al., 2024) and BatchTopK (Bussmann et al., 2024a) that enforce a fixed number of active features (k), or penalty-based methods like Gated SAEs (Rajamanoharan et al., 2024a) and P-anneal (Karvonen et al., 2024) that apply consistent regularization pressure, these designs create fundamental inefficiencies where conceptually simple inputs receive excessive representational capacity while complex inputs face insufficient feature allocation. This limitation becomes increasingly problematic at scale as Gao et al. (2024) demonstrate that larger language models require proportionally more features to achieve comparable reconstruction quality. Moreover, finding optimal sparsity settings requires extensive hyperparameter experimentation to navigate the critical reconstruction-sparsity trade-off (Karvonen et al., 2025).

To overcome this limitation, we propose that *a sparse autoencoder should adaptively adjusts sparsity levels based on input complexity*. When a simple semantic concept can be effectively explained

and reconstructed using only a few features, activating additional features becomes unnecessary. This approach not only conserves computational resources but also prevents unnecessary feature activation on simpler texts, reducing both overfitting and representational noise.

How can we define semantic simplicity or complexity within LLM representation spaces? Peters et al. (2018) demonstrated that probes can map LLM intermediate representations to semantic and syntactic information. Similarly, higher-level concepts such as political perspective (Kim et al., 2025), sentiment (Tigges et al., 2023), and spatiotemporal information (Gurnee & Tegmark, 2023) have been shown to be linearly represented in activation spaces. Based on these observations, we hypothesize that the internal representations formed by large language models during text processing naturally encode multidimensional properties of text, including its complexity.

Our study first establishes that text complexity is linearly encoded in language model representations. We score contexts using GPT-4.1-mini API (OpenAI, 2024) across six semantic dimensions to create aggregate complexity scores, then train linear regression probes on model activations to predict these scores. Experiments with eight different scale LLMs demonstrate high correlation coefficient, confirming our hypothesis that LLMs naturally encode text complexity in their representation spaces. Analysis reveals that texts of varying complexity require proportional representational capacity, with complex inputs necessitating more active features for accurate encoding. This key insight suggests that adaptive sparsity mechanisms could significantly improve autoencoder efficiency.

Based on our complexity prediction capabilities, we develop the **Adaptive Top K Sparse Autoencoder (AdaptiveK SAE)**, which to our knowledge is the first work to solve computational efficiency bottlenecks in sparse autoencoder training while maintaining feature quality. Our approach quantifies context complexity using a linear probe trained on multi-dimensional complexity annotations. This score determines an appropriate sparsity level, activating more features for complex inputs while maintaining high sparsity for simpler ones. This complexity-driven adaptation better balances reconstruction quality, sparsity, and interpretability. Experiments demonstrate our framework outperforms fixed-sparsity approaches across multiple model scales. Code is available at: *https://anonymous.4open.science/r/adaptiveK-5258*. Our main contributions can be summarized as:

- We propose AdaptiveK Sparse Autoencoders, a novel framework that dynamically adjusts sparsity levels based on input complexity.
- We demonstrate that text complexity is linearly encoded in model representations, establishing a direct relationship between semantic complexity and representational capacity needs in LLMs.
- Experiments across ten LLMs show that our SAE consistently outperforms fixed-sparsity baselines on reconstruction fidelity, explained variance, cosine similarity and other metrics.

## 2 PRELIMINARIES AND MOTIVATION

### 2.1 BASELINE SPARSE AUTOENCODER

Following the initial works that introduced SAEs for decomposing model representations (Cunningham et al., 2023), a variety of architectural refinements have emerged. Our comparative analysis of these baselines was facilitated by the dictionary_learning library (Marks et al., 2024b). Foundational ReLU SAEs (Bricken et al., 2023) and the refined one (Anthropic Interpretability Team, 2024) typically map an input activation $x \in \mathbb{R}^d$ to a sparse latent $z \in \mathbb{R}^M$ (where $M \gg d$) and then to a reconstruction $\hat{x}$. The core operations involve an encoder:

$$z = \text{ReLU}(W_{enc}(x - b_{pre}) + b_{enc}), \tag{1}$$

and a decoder:

$$\hat{x} = W_{dec}z + b_{pre}, \tag{2}$$

with a training loss that combines reconstruction error with an $L_1$ sparsity penalty on $z$.

$$L = \|x - \hat{x}\|_2^2 + \lambda\|z\|_1. \tag{3}$$

To mitigate issues like feature shrinkage from the $L_1$ penalty, Gated SAEs (Rajamanoharan et al., 2024a) decouple the $L_1$-penalized feature selection gate $g(x)$ from magnitude estimation $m(x)$, forming activations as $z = g(x) \odot m(x)$. Other approaches enforce sparsity directly: TopK SAEs (Gao et al., 2024) select the K highest pre-activations.

$$z = \text{ReLU}(\text{ReLU}(W_{enc}(x - b_{pre}) + b_{enc}), K). \tag{4}$$

BatchTopK SAEs (Bussmann et al., 2024a) extend this by selecting the top $N \times K$ activations across a batch of $N$ samples. JumpReLU SAEs (Rajamanoharan et al., 2024b) utilize a discontinuous activation $z_i = \text{act}_i \cdot H(\text{act}_i - \theta_i)$ with a learned threshold $\theta_i$ (where $\text{act}_i$ is preactivation and $H$ is Heaviside), often paired with a $L_0$ sparsity term and trained via Straight-Through Estimators. For refining loss-based sparsity, P-anneal ReLU SAEs (Karvonen et al., 2024) employ an $L_p$ norm penalty, $\lambda \sum_i |z_i|^p$, where $p$ anneals from 1 towards 0. Lastly, to address feature hierarchy and issues like splitting or absorption, Matryoshka BatchTopK SAEs (Bussmann et al., 2024b) train nested dictionaries of increasing capacity, using BatchTopK for sparsity within each level.

## 2.2 Our Motivation

Despite advances in SAE scalability, current approaches face a limitation: they apply uniform sparsity constraints regardless of input complexity. This "one-size-fits-all" approach creates inefficiencies across the representation space. Whether employing fixed activation methods such as TopK (Gao et al., 2024) or regularization techniques like Gated SAEs (Rajamanoharan et al., 2024a) existing architectures cannot adapt to the varying complexity demands of different inputs. As shown in Figure 1 conventional SAE methods with fixed K maintain constant activation (*e.g.*, 80 features across all complexities), while our AdaptiveK dynami-

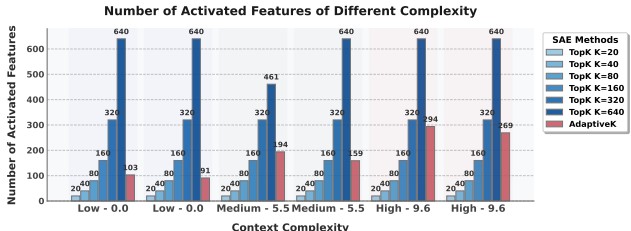

Figure 1: Two samples were selected from each complexity level (simplest=0, moderate=5.5, and most complex=9.6) from test set. In TopK SAE, feature activation strictly follows the k value between 20-320, but often falls below the threshold when k=640. For Pythia-160M, fixed TopK SAEs (blue) maintain constant activation, while AdaptiveK (red) dynamically scales with text complexity.

cally scales from 103 to 394 features based on input complexity. This inefficiency becomes more pronounced at scale, determining optimal sparsity parameters necessitates extensive hyperparameter optimization to balance reconstruction fidelity against sparsity constraints.

Our approach is motivated by the observation that text complexity is linearly encoded in language model representations, suggesting a more efficient solution: adaptively adjusting sparsity levels based on input complexity. Simple inputs require fewer features for reconstruction, while complex ones need more representational capacity. This adaptive allocation improves computational efficiency, reduces overfitting on simple inputs, and enhances interpretability, all achieved within a single training run without extensive hyperparameter tuning.

# 3 The Proposed AdaptiveK Sparse Autoencoder

In this work, we propose AdaptiveK Sparse Autoencoder that dynamically adjusts sparsity to input complexity. By allocating representational capacity in proportion to content complexity, AdaptiveK addresses the core inefficiency of uniform sparsity (Figure 2). The architecture comprises two components: (1) a linear probe that predicts input complexity (Section 3.1); and (2) an SAE with AdaptiveK activation (Section 3.2). We also present a three-phase training procedure that stabilizes and improves learning (Section 3.3).

## 3.1 Training Linear Probes to Predict Context Complexity

### 3.1.1 Linear Probe for Complexity Prediction

Our dataset comprises contexts from pile-uncopyrighted (Gao et al., 2020), each containing 1024 tokens. Complexity is quantified on a scale from 0 to 10, based on a six-dimensional evaluation (lexical complexity, syntactic complexity, conceptual density, domain specificity, logical structure and logical structure) by GPT-4.1-mini (scoring prompts detailed in Appendix B), yielding target labels $y_i$. These complexity scores are floating-point numbers with one decimal place. For each

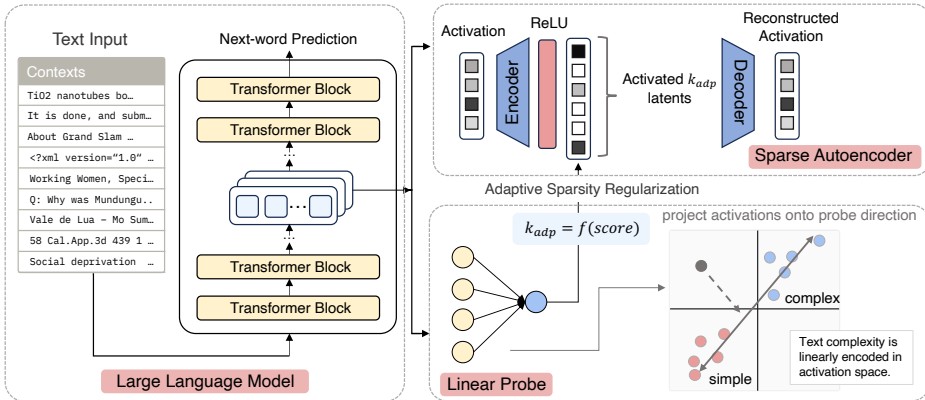

Figure 2: Overall pipeline of the AdaptiveK SAE. Input text is fed into a LLM to extract internal activations, which are then passed through both a linear probe that predicts text complexity and a SAE for decomposition. During training, the linear probe's complexity score dynamically determines the number of features to activate, allowing more features for complex inputs and fewer for simple ones.

context, we extract the hidden state activations of the last token from selected layers of the autoregressive transformer language models Pythia (70M, 160M) (Biderman et al., 2023), Gemma-2 (2B, 9B) (Team et al., 2024), Llama-3.1 (8B) (Grattafiori et al., 2024), Qwen-3 (8B, 14B) (Yang et al., 2025) and Phi-4 (14B) (Abdin et al., 2024), which encapsulate contextual information. The input data for each context $i$ is represented as a pair $[x_i, y_i]$, where $x_i \in \mathbb{R}^{d_{\text{model}}}$ is the vector of hidden state activations and $y_i \in \mathbb{R}$ is the corresponding complexity score.

Following Gurnee & Tegmark (2023), we employ ridge regression to mitigate overfitting and multicollinearity issues common with high-dimensional activation vectors (Kim et al., 2025). The objective is to find the weight vector $w \in \mathbb{R}^{d_{\text{model}}}$ and bias term $b \in \mathbb{R}$ that minimize the $L_2$-regularized squared loss:

$$L(w, b) = \frac{1}{n} \sum_{i=1}^{n} (y_i - (w^T x_i + b))^2 + \frac{\lambda}{2} ||w||_2^2,$$ (5)

where $n$ is the number of training contexts, and $\lambda$ is the regularization hyperparameter. The activation matrix $A \in \mathbb{R}^{n \times d_{\text{model}}}$ is constructed by concatenating these feature vectors, with the target defined as $y \in \mathbb{R}^n$. With implicit bias handling (*e.g.*, by centering data or adding a feature column of ones to $A$), the optimal weight vector $\hat{w}$ is given by the closed-form solution (Belinkov, 2022):

$$\hat{w} = \left( A^T A + \lambda I \right)^{-1} A^T y.$$ (6)

To determine the optimal regularization strength $\lambda$, we perform 5-fold cross-validation. For each $\lambda$ in the set $\{0.001, 0.01, 0.1, 1.0, 10.0, 100.0, 1000.0\}$, the probe is trained on four folds and evaluated on the held-out fold using the root mean squared error (RMSE). We select $\lambda = 100.0$, which yields the lowest average RMSE across the folds. The final probe, with parameters $(\hat{w}, \hat{b})$, is then trained on the entire dataset using this optimal $\lambda$. Subsequently, this trained probe is used to predict complexity scores for new contexts based on their last token activation $x$ via the linear function $\hat{y} = \hat{w}^T x + \hat{b}$. More training details will be presented in Appendix B.

### 3.1.2 EVALUATING THE LINEAR REPRESENTATION HYPOTHESIS

As will be mentioned in Appendix A.2 , many high-level features have been demonstrated to exist linearly in LLM representation spaces. However, context complexity is indeed multifaceted, spanning lexical, syntactic, and other linguistic dimensions, and thus differs from the single-attribute features studied in prior work.

By comparing the performance of linear probes, MLP, and XGBoost in predicting context complexity, we provide evidence for the linear encoding of context complexity features in language model representation spaces. We contrasted a single-hidden-layer MLP with the structure $f(\mathbf{x}) = \mathbf{W}_2 \text{ReLU}(\mathbf{W}_1 \mathbf{x} + \mathbf{b}_1) + \mathbf{b}_2$ containing 256 neurons in the hidden layer, and an XGBoost model that minimizes error through gradient boosting across multiple decision trees, with the prediction formula $\hat{y}_i = \sum_{k=1}^{K} f_k(\mathbf{x}_i)$, against our linear regression probe.

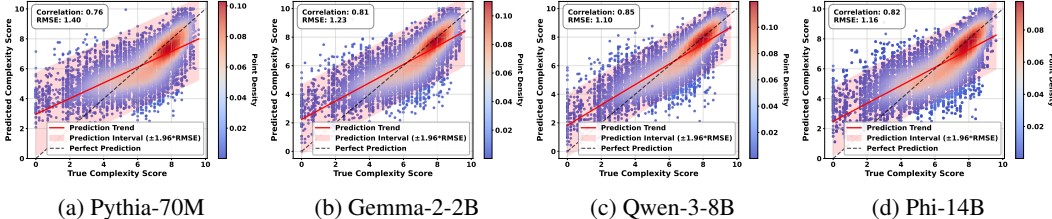

| (a) Pythia-70M | (b) Gemma-2-2B | (c) Qwen-3-8B | (d) Phi-14B |

Figure 3: Visualization of linear probe performance across different LLM scales. Points represent test contexts, with redder areas indicating higher sample density. The red line depicts predicted complexity trends. Most samples fall within prediction intervals, confirming the linear probe's effectiveness. Spearman Correlation and RMSE values (upper left) demonstrate improved prediction accuracy with increasing model scale. More LLM results are in Figure 10.

As shown in Table 1, linear model is comparable to theoretically more expressive nonlinear models like XGBoost across all three metrics: Pearson and Spearman correlations and RMSE. These findings extend the applicability of the linear representation hypothesis to multifaceted features such as context complexity. Figure 3 visualizes our probe's performance on test data across eight models. Each point represents one context from the test set, showing the predicted versus true complexity score. With most points falling within the prediction interval ($\pm 1.96 \times$ RMSE), this performance confirms that context complexity is largely encoded linearly in representation space.

Table 1: Comparison of Linear Probe, MLP, and XGBoost models for context complexity prediction on Pythia-70M, trained on 250,000 contexts and evaluated on 10,000 test samples using RMSE, Pearson, and Spearman metrics.

| Probe | RMSE | Pearson | Spearman |
|---|---|---|---|
| Linear | 1.41 | 0.72 | 0.76 |
| MLP | 1.37 | 0.74 | 0.77 |
| XGBoost | 1.42 | 0.71 | 0.74 |

## 3.2 ADAPTIVEK SPARSE AUTOENCODER ARCHITECTURE

Unlike existing SAEs apply uniform sparsity constraints across all inputs, requiring extensive hyperparameter experimentation, our AdaptiveK Sparse Autoencoder incorporates a complexity estimation component that adaptively determines the appropriate sparsity level for each context. The overall pipeline is shown in Figure 2.

For an input activation vector $x \in \mathbb{R}^d$, we compute a complexity score using the linear probe from Section 3.1: $c = \hat{w}^T x + \hat{b}$, where $\hat{w} \in \mathbb{R}^d$ and $\hat{b} \in \mathbb{R}$ are trained from the ridge regression. This score is then mapped to a sparsity level $k_{\text{adp}}$ through a sigmoid-based transformation:

$$k_{\text{adp}} = k_{\text{min}} + \frac{1}{1 + e^{-s\left(\frac{c - c_{\text{min}}}{c_{\text{max}} - c_{\text{min}}} - 0.5\right)}} \left(k_{\text{max}} - k_{\text{min}}\right). \tag{7}$$

where $k_{\text{min}}$ and $k_{\text{max}}$ define the range of possible $k_{\text{adp}}$ values, $c_{\text{min}}$ and $c_{\text{max}}$ represent the minimum and maximum complexity scores, and $s$ controls the steepness of the sigmoid function. The sparse autoencoder component then processes the input with an adaptive TopK activation function:

$$z = \text{TopK}\left(W_{\text{enc}}\left(x - b_{\text{pre}}\right), k_{\text{adp}}\right), \tag{8}$$

where $\text{TopK}(\cdot, k_{\text{adp}})$ retains only the $k_{\text{adp}}$ largest activations and sets all others to zero. The encoder matrix $W_{\text{enc}} \in \mathbb{R}^{M \times d}$, decoder matrix $W_{\text{dec}} \in \mathbb{R}^{d \times M}$, and bias vector $b_{\text{pre}} \in \mathbb{R}^d$ are trainable parameters. The output $\hat{x}$ is given followed Equation 2.

Therefore, the AdaptiveK SAE eliminates the computational burden of training separate models for each sparsity level to find the optimal trade-off, requiring only a single training run. Additionally, it addresses the feature suppression problem that occurs with L1 penalties while improving performance on context-level tasks by allocating representational capacity proportional to context complexity. Experiments in Section 4 demonstrate that this complexity-driven adaptation achieves better reconstruction without requiring extensive hyperparameter tuning.

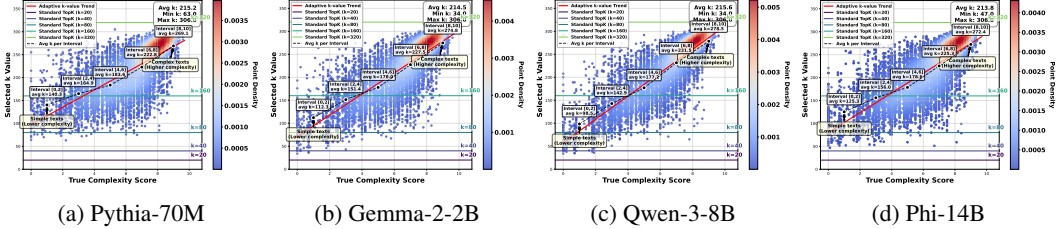

| (a) Pythia-70M | (b) Gemma-2-2B | (c) Qwen-3-8B | (d) Phi-14B |

Figure 4: Visualization of Dynamic Feature Allocation by Text Complexity showing the relationship between complexity scores and allocated feature counts (K values). Average K values per complexity interval (connected by red lines) demonstrate that complex texts receive higher K allocations, with this relationship becoming increasingly linear as LLM scale grows. Horizontal lines indicate fixed Standard TopK baselines with K values on the right. More LLM results are in Figure 11.

### 3.3 ADAPTIVEK SPARSE AUTOENCODER TRAINING

We employ a three-phase training for AdaptiveK SAE (see Algorithm 1). First, we pretrain the complexity estimation probe as described in Section 3.1.

In the second phase, we freeze the probe parameters and train only the SAE components using:

$$L_{\text{SAE}} = L_{\text{recon}} + \alpha L_{\text{sparsity}} + \beta L_{\text{aux}}, \quad (9)$$

where $L_{\text{recon}} = |x - \hat{x}|_2^2$ is the reconstruction loss, $L_{\text{sparsity}} = \frac{\|z\|_1}{\|x\|_2}$ is the normalized $L_1$ penalty with weight $\alpha = 0.005$, and $L_{\text{aux}}$ is the auxiliary loss with weight $\beta = 1/32$ for reactivating dead features. We set $base\_k = 80$, $min\_k = 20$, and $max\_k = 320$ for the sparsity range.

In the final joint fine-tuning phase, we jointly optimize both components with:

$$L_{\text{joint}} = L_{\text{SAE}} + \gamma(L_{\text{probe}} + \delta L_{\text{deviation}}), \quad (10)$$

where $\gamma = 0.9$ controls the probe loss weight and $L_{\text{deviation}} = |w - w^0|_2 + |b - b^0|$ penalizes deviations from pretrained parameters, initially with $\delta = 0.2$. This penalty prevents the SAE's reconstruction objective from corrupting the probe's complexity mapping. We adaptively adjust $\delta$ between 0.01 and 0.5, decreasing $\delta$ when probe loss improves and increasing $\delta$ when it stagnates.

Our implementation uses a AdaptiveKBuffer that extracts last-token representations from contexts (see Section 3.1.1 for details), reducing memory usage while tracking complexity scores for balanced training. Adam optimizer (Kingma & Ba, 2014) is applied with learning rate 1e-3, warm-up over 15 steps, and linear decay starting at 70% of training. More SAE training details are in Appendix C.

---

**Algorithm 1:** AdaptiveK SAE Training

**Input:** Activation data $D = \{x_i\}_{i=1}^N$, random initialized SAE.

1 // Phase 1: Complexity probe pretraining
2 Train linear probe to predict complexity scores from activations

// Phase 2: SAE training with frozen probe
3 // Phase 2: SAE training with frozen probe
4 **while** *not converged* **do**
5    Apply adaptive sparsity constraints based on complexity
6    Update SAE parameters using $L_{SAE} = L_{recon} + \alpha L_{sparsity} + \beta L_{aux}$

7 // Phase 3: Joint fine-tuning
8 **while** *not converged* **do**
9    Update all parameters using $L_{joint} = L_{SAE} + \gamma(L_{probe} + \delta L_{deviation})$

**Output:** Trained AdaptiveK SAE.

---

## 4 EXPERIMENTS

In this section, we evaluate our AdaptiveK to answer the following research questions (RQs):

- **RQ1:** How does the relationship between text complexity and adaptive k-values manifest across different language model scales?
- **RQ2:** To what extent does our adaptive sparsity mechanism improve reconstruction quality metrics (L2 loss, variance explained, and cosine similarity) compared to fixed-sparsity approaches?
- **RQ3:** How does AdaptiveK SAE's performance on the Pareto frontier balance the trade-off between sparsity and reconstruction fidelity compared to baseline methods?

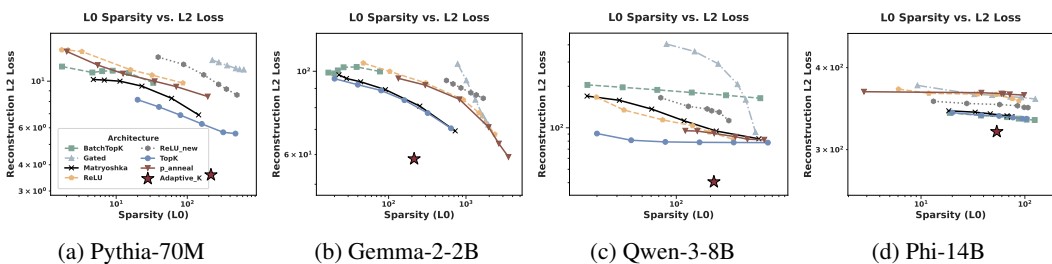

(a) Pythia-70M      (b) Gemma-2-2B      (c) Qwen-3-8B      (d) Phi-14B

Figure 5: L2 Loss pareto frontier results. More LLM results are in Figure 12.

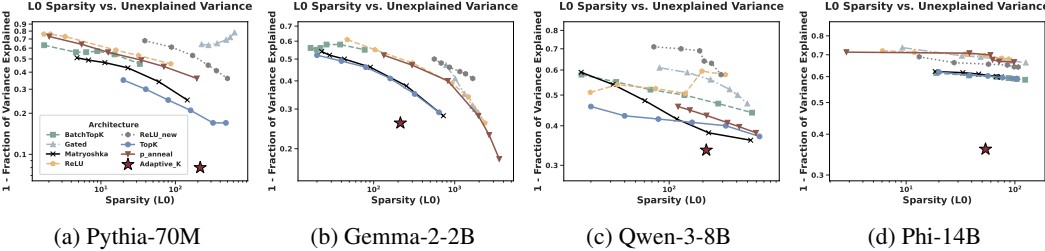

(a) Pythia-70M      (b) Gemma-2-2B      (c) Qwen-3-8B      (d) Phi-14B

Figure 6: Unexplained Variance pareto frontier results. More LLM results are in Figure 13.

- **RQ4:** How is LLM interpretability defined, and how can the interpretability of AdaptiveK SAE be measured at a human-understandable level?

## 4.1 EXPERIMENTAL SETTINGS

We train SAEs on 8 language models of increasing scale: Pythia (70M, 160M), Gemma-2 (2B, 9B), Llama-3.1 (8B), Qwen-3 (8B, 14B) and Phi-4 (14B), with hidden dimensions from 512 to 5120. All trained SAEs have a dictionary size of 16384, i.e., 16k latents. Unlike token-level approaches, we operate at context level (though we demonstrate in Section H that token-level evaluation is also applicable) using pile-uncopyrighted data (Gao et al., 2020): 250,000 training contexts and 10,000 test contexts, each with 2048 tokens. We train on the last token representation of each context, which efficiently captures accumulated contextual information for complexity-driven sparsity adaptation.

For experimental evaluation, we compare our AdaptiveK SAE against 7 SAE baselines: (1) ReLU SAEs (Bricken et al., 2023), using ReLU activation with L1 penalty; (2) refined ReLU_new SAEs (Anthropic Interpretability Team, 2024); (3) TopK SAEs (Gao et al., 2024), which select the K highest activations; (4) BatchTopK SAEs (Bussmann et al., 2024a), extending TopK across batches; (5) Gated SAEs (Rajamanoharan et al., 2024a), decoupling feature selection from magnitude estimation; (6) P-anneal SAEs (Karvonen et al., 2024), using annealing $L_p$ norm penalties; and (7) Matryoshka SAEs (Bussmann et al., 2024b), training nested dictionaries with increasing capacity.

Beyond the primary metrics, we evaluated AdaptiveK using SAEBench metrics (Karvonen et al., 2025) (Appendix F.3). Additional analyses include layer-wise performance (Appendix F.1), extensions to encoder-only and encoder-decoder models (Appendix F.2), hyperparameter analysis (Appendix F.4), and training efficiency analysis (Appendix G).

## 4.2 RELATIONSHIP BETWEEN COMPLEXITY AND k-VALUES

We plotted the relationship between true complexity and k-value selection on the test set, calculated the average k-value for each complexity interval, and marked the fixed k-values that TopK would select. As shown in Figure 4, there is an approximately linear relationship between text complexity and allocated k-values, which becomes increasingly evident as model scale increases.

On one hand, Figure 3 reveals that both predicted complexity and activated feature count increase with sample complexity, validating that complex texts correctly receive larger k values and activate more features. On the other hand, Figure 5 indicate that that AdaptiveK consistently achieves lower reconstruction errors than TopK SAE across all k-values, demonstrating the benefit of adaptive resource allocation by using fewer features for simple texts and more for complex ones. Additional probe evaluations using PCA and layer-wise analysis are in Appendices E.1 and E.2.

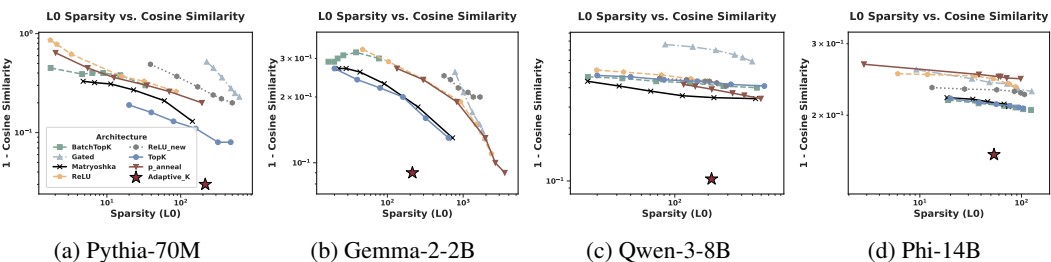

(a) Pythia-70M  (b) Gemma-2-2B  (c) Qwen-3-8B  (d) Phi-14B

Figure 7: Cosine Similarity pareto frontier results. More LLM results are in Figure 14.

## 4.3 PARETO FRONTIER RESULTS

We evaluated three different metrics vs. sparsity frontier for benchmark SAEs with different sparsity constraints. All SAEs have a dictionary size of 16384. These three metrics are:

- **Reconstruction L2 Loss:** $\|x - \hat{x}\|_2^2$ averages the squared Euclidean distance between each reconstruction and its input, lower is better.
- **Fraction of Variance Explained:** $\frac{\text{Var}(x-\hat{x})}{\text{Var}(x)}$ measures how much input variability is captured by the reconstruction. The plots show 1 minus this value (unexplained variance, lower is better).
- **Cosine Similarity:** $\frac{x \cdot \hat{x}}{\|x\|_2 \|\hat{x}\|_2}$ evaluates directional fidelity between original and reconstructed vectors. The plots show 1 minus this value, thus lower is better.

Figures 5, 6, and 7 show AdaptiveK consistently outperforms all other SAEs across different LLMs in reconstruction error, cosine similarity, and explained variance at equivalent sparsity. While some SAEs match or exceed these metrics at extremely high sparsity (over $10\times$ greater), such cases violate the sparsity-fidelity tradeoff since infinite-width SAE can theoretically reconstruct perfectly (Karvonen et al., 2025). Thus, AdaptiveK transcends the traditional Pareto Frontier, achieving results unreachable by other SAEs regardless of parameter tuning.

Notably, across all model scales, AdaptiveK's reconstructed activations more accurately match the originals in both distance and direction (Figure 5, 12, 7, 14) while explaining a higher proportion of data variance (Figure 6, 13), demonstrating AdaptiveK's robustness and generalizability.

## 4.4 INTERPRETABILITY ANALYSIS OF ADAPTIVEK SAE

### 4.4.1 DEFINITION OF LLM INTERPRETABILITY

Following established mechanistic interpretability research (Rai et al., 2025; Shu et al., 2025; Zhao et al., 2023), we define LLM interpretability as how well learned features correspond to semantically coherent, human-understandable concepts that can be consistently identified across different contexts. Specifically, we consider a feature to be interpretable if it satisfies: (1) input-based semantic coherence: it activates on inputs sharing a clear conceptual relationship, and (2) output-based human comprehensibility: it influences semantically consistent vocabulary prediction.

### 4.4.2 MEASURES OF INTERPRETABILITY

For input-based analysis, we implemented the MaxAct (Bricken et al., 2023) method to identify text segments that maximally activate each SAE feature. Specifically, for each target feature $w_m$, we fed corpus texts $x$ from our test set into the original language model to obtain hidden representations $\mathbf{h}_x = f_{<l}(x)$ at layer $l$, computed feature activation strengths $a_m(x) = \text{ReLU}\left(\mathbf{w}_m^\top \cdot \mathbf{h}_x + b_m\right)$ through the SAE encoder, and selected the top-$K$ text segments with highest activations:

$$\mathcal{I}_m = \arg\max_{|\mathcal{X}'|=K} \sum_{x \in \mathcal{X}'} a_m(x) \tag{11}$$

Through experiments on Gemma-2-9B comparing against TopK SAE (k=80), we analyzed thematic consistency of high-activation texts to quantify whether features learned monosemantic, human-interpretable concepts. Figure 8 shows that for biomedical texts, AdaptiveK's activation patterns focus precisely on core professional concepts, including technical terms ("resonance spectroscopy," "MRS," "ppm"), biochemical concepts ("acetylcarnitine," "peak"), and methodological vocabulary

| ◆ *AdaptiveK SAE* | ◆ *TopK(K=80) SAE* |
|---|---|
| vivo proton magnetic resonance spectroscopy (1H-MRS), a new peak resonating at 2.13 ppm post-exercise has been attributed in the literature to the acetyl groups of acetylcarnitine. Since this peak is inconsistently generated by various submaximal exercise regimens, this study aimed at (a) verification of the previous chemical assignment, (b) determination of exercise conditions necessary for its induction, and (c) documentation of the recovery kinetics through 60 minutes following exercise. Ten... | vivo proton magnetic resonance spectroscopy (1H-MRS), a new peak resonating at 2.13 ppm post-exercise has been attributed in the literature to the acetyl groups of acetylcarnitine. Since this peak is inconsistently generated by various submaximal exercise regimens, this study aimed at (a) verification of the previous necessary for its induction, and (c) documentation of the recovery kinetics through 60 minutes following exercise. Ten... |

Figure 8: Input-based interpretability analysis using MaxAct method.

| | *Gemma-2-9B* | | | *Llama-3.1-8B* | | | |
|---|---|---|---|---|---|---|---|
| **Feature 5637** | | **Feature 2370** | | **Feature 2865** | | **Feature 8949** | |
| positive tokens | negative tokens | positive tokens | negative tokens | positive tokens | negative tokens | positive tokens | negative tokens |
| "getTitle": 0.38 | "Ken": -0.21 | "abstract": 0.33 | "تحدى": -0.19 | "micro": 0.62 | "فإن": -0.34 | "expansion": 0.32 | "LookAnd": -0.21 |
| "Title": 0.36 | "Cat": -0.21 | "abstra": 0.33 | "shoot": -0.18 | "Micro": 0.58 | "unsafe": -0.32 | "Expansion": 0.32 | "UserScript": -0.20 |
| "TITLE": 0.35 | "climat": -0.21 | "abstract": 0.32 | "}{*}{}": -0.18 | "MICRO": 0.46 | "erratic": -0.31 | "expansions": 0.31 | "وتسجيلات": -0.20 |
| "Titles": 0.34 | "Turk": -0.20 | "Abstract": 0.30 | "qualitatively": -0.18 | "Macro": 0.41 | "({\"": -0.31 | "Expansion": 0.31 | "surla": -0.19 |
| "title": 0.34 | "baba": -0.20 | "ABSTRACT": 0.29 | "mukana": -0.18 | "microscopic": 0.40 | "eruption": -0.31 | "expand": 0.30 | "KommentareTeilen": -0.19 |
| "titles": 0.34 | "shepherds": -0.20 | "Abstract": 0.29 | "endpush": -0.18 | "micro": 0.39 | "respective": -0.30 | "expansion": 0.30 | "Personensuche": -0.19 |
| "getTitle": 0.33 | "shepherd": -0.20 | "abstracto": 0.29 | "taux": -0.17 | "minuscule": 0.39 | "_worker": -0.30 | "expand": 0.30 | "WebServlet": -0.19 |
| "Title": 0.32 | "Cong": -0.20 | "ABSTRACT": 0.28 | "shoots": -0.17 | "Milo": 0.39 | "LEX": -0.30 | "Expand": 0.29 | "CloseOperation": -0.18 |
| "Titel": 0.32 | "Union": -0.20 | "abstracta": 0.28 | "hábito": -0.17 | "Micro": 0.38 | "——": -0.30 | "expanding": 0.29 | "PerformLayout": -0.18 |
| "title": 0.31 | "She": -0.20 | "abstracted": 0.27 | "collaborators": -0.17 | ".micro": 0.38 | "Preis": -0.29 | "Expanding": 0.29 | "աշխատություններ": -0.18 |

Figure 9: Output-based interpretability analysis using VocabProj method.

("verification," "chemical," "induction"). In contrast, TopK activates both similar professional terminology and numerous semantically weak function words ("been," "groups," "generated," "aimed"), diluting feature focus and inefficient computational resource utilization with diminishing semantic returns despite employing more features. This demonstrates AdaptiveK's ability to identify semantic complexity and allocate appropriate high-quality features while avoiding irrelevant activations.

For output-based analysis, we used VocabProj (Shu et al., 2025) to analyze feature influence patterns on model vocabulary prediction. By computing inner products between each feature vector $w_m$ in the SAE decoder and the language model's output vocabulary embedding matrix $f_{out}(w)$, we obtained logits quantify which words model tends to generate or suppress when features activate:

$$\mathcal{I}_m = \arg\max_{w \in \mathcal{V}} f_{out}(w) \cdot \mathbf{w}_m^\top \tag{12}$$

We extracted the top 10 positive and negative associated vocabulary items per feature to evaluate semantic coherence across Llama-3.1-8B and Gemma-2-9B (Figure 9). In Gemma-2-9B, Feature 5637 specializes in "title" concepts, with positive vocabulary centered around title variants including different capitalizations and languages (*e.g.*, German "Titel"), while negative vocabulary contains unrelated noun. Feature 2370 focuses on "abstract" concepts, similarly demonstrating cross-linguistic and cross-format unity with variants like Spanish "abstracto", and negative vocabulary containing technical symbols and irrelevant words. In Llama-3.1-8B, Feature 2865 concentrates on "microscopic" concepts, with positive vocabulary extending from "micro" to related terms like "microscopic" and "minuscule," showing semantic expansion from root to concept. Feature 8949 captures "expansion" across tenses and noun forms, reflecting complete semantic field coverage.

These results reveal AdaptiveK's capability for learning semantically coherent features capturing not only literal expressions but also concept variants across contexts, languages, and grammatical forms. The negative vocabulary ranging from foreign terms to technical terminology consistently forms clear semantic contrasts with target concepts, indicating robust conceptual boundary identification.

## 5 CONCLUSION

In this paper, we introduce AdaptiveK SAE, which demonstrates that dynamically adjusting sparsity based on input complexity significantly improves representation decomposition in LLMs. By establishing that text complexity is linearly encoded in LLM activations, we developed a framework that allocates computational resources proportionally to content complexity, eliminating extensive hyperparameter tuning while consistently outperforming fixed-sparsity baselines across multiple model scales. Experiments on 8 LLMs confirm that complexity-driven adaptation achieves better reconstruction fidelity, explained variance, cosine similarity and interpretability.

## REPRODUCIBILITY STATEMENT

We provide comprehensive resources to ensure the reproducibility of our work. Our code implementation, including the complete AdaptiveK SAE framework, linear probe training pipeline, and evaluation scripts, is available at *https://anonymous.4open.science/r/adaptiveK-5258*. The training dataset consists of 250,000 contexts from pile-uncopyrighted (Gao et al., 2020), with complexity annotations generated using GPT-4.1-mini based on six semantic dimensions detailed in Appendix B. All training details and hyperparameters are specified in Section 3.3 and Appendix F.4. We trained on the last token representations of 1024-token contexts using Adam optimizer with learning rate 1e-3. The three-phase training procedure and adaptive deviation penalty mechanism are fully described in Algorithm 1. All baseline SAE implementations were obtained from the dictionary_learning library (Marks et al., 2024b), with their specific configurations listed in Table 2.

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

## A    RELATED WORK

### A.1    SPARSE AUTOENCODERS AND IMPROVEMENTS

A significant challenge in neural network interpretability is polysemanticity, where units (*e.g.*, neurons) respond to diverse, semantically distinct inputs, complicating functional analysis (Elhage et al., 2022; Olah et al., 2020). The superposition hypothesis (Park et al., 2023) suggests that networks represent more features than available neurons by encoding them as directions in activation space, rather than solely via individual neuron activities. SAEs offer an unsupervised dictionary learning approach to tackle this, decomposing internal network activations (particularly in LLMs) to reveal latent interpretable units (Ferrando et al., 2024; Shu et al., 2025). Core to SAEs is an encoder mapping an activation $x$ to a higher-dimensional sparse representation $z$, and a decoder reconstructing $\hat{x}$ from $z$. Goal for SAEs are to isolate monosemantic and composable features, thus offering a more faithful representation of the model's internal computational state (Huben et al., 2023).

Following initial SAE proposals (Cunningham et al., 2023), research rapidly advanced SAE design. Efforts addressed limitations like L1 penalty-induced shrinkage, leading to Gated SAEs (Rajamanoharan et al., 2024a). Alternative sparsity mechanisms emerged, including TopK (Gao et al., 2024), BatchTopK (Bussmann et al., 2024a), JumpReLU (Rajamanoharan et al., 2024b), and ProLU (Taggart, 2024). Architectural innovations like Switch SAEs improved computational scaling (Mudide et al., 2024), while Matryoshka SAEs targeted feature hierarchy and splitting/absorption issues (Bussmann et al., 2024b). Optimization objectives were also refined through techniques like P-annealing (Karvonen et al., 2024), feature alignment (Marks et al., 2024a), and end-to-end training (Braun et al., 2024). While these advancements often optimize proxy metrics like sparsity and fidelity, their alignment with true interpretability remains an active area of evaluation.

### A.2    LINEAR PROBE IN LARGE LANGUAGE MODELS

Linear probes have emerged as a fundamental method for elucidating how LLMs represent complex information within their activation spaces (Li et al., 2023; Von Rütte et al., 2024; Mikolov et al., 2013). This approach is grounded in the hypothesis that important high-level concepts are encoded linearly as directions in representational space (Kim et al., 2025; Liu et al., 2024). A substantial body of research supports the finding that linear probes are more effective than nonlinear probes at aligning model representations with specific behaviors. For instance, Gurnee & Tegmark (2023) applied linear probes to Llama-2 models to reveal that LLMs learn linear representations of space and time that are robust to prompting variations, unified across entity types, and encoded by specific neurons within the network. Similarly, Tigges et al. (2023) demonstrated that sentiment in language models emerges along specific linear directions in activation space, with a single dimension causally controlling sentiment polarity (positive versus negative) in model outputs through direct interventions. Additionally, concepts such as topic direction (Turner et al., 2023), political ideology (Kim et al., 2025), game states (Nanda et al., 2023), truthfulness (Marks & Tegmark, 2023) and safety (Arditi et al., 2024) have also been identified as important features that are linearly encoded within the internal activation spaces of LLMs.

## B    LINEAR PROBES TRAINING DETAILS

Our training methodology employs texts from the pile-uncopyrighted corpus, which are processed through a tokenizer and aggregated into contexts of 1024 tokens each. Each context undergoes a comprehensive six-dimensional evaluation by GPT-4.1-mini, resulting in a normalized complexity score between 0 and 10 with one decimal place precision. Our dataset comprises 250,000 training contexts and 10,000 test contexts.

For each context, we extract the activation vector of the final token as the representational vector for that context, with dimensionality matching that of the model's hidden layer. During training, batches of unprocessed activation values are sequentially retrieved from the buffer and marked as processed. When all activations have been utilized, the buffer is replenished and shuffled to introduce stochasticity. This iterative cycle continues until a sufficient quantity of activation samples has been accumulated for effective model training.

Below are some examples with various complexity scores, their predicted complexities by the linear probe, corresponding K-values, and the number of activated latent features through AdaptiveK SAE:

---

**Sample 1: True Complexity: 1.2, Pred Complexity: 3.18, K value: 95, Activated Features: 95**

```
\u00f4m\u00e9\",\n          \"Employ\u00e9 de commerce\",\n          \"Employ\u00e9 de
    commerce CFC\",\n          \"Employ\u00e9 de remont\u00e9es m\u00e9caniques AFP\",\n
            \"Employ\u00e9 d\u2019exploitation AFP\",\n          \"Employ\u00e9 en
    cuisine AFP\",\n          \"Employ\u00e9 en h\u00f4tellerie AFP\",\n          \"Employ
    \u00e9 en industrie laiti\u00e8re AFP\",\n          \"Employ\u00e9 en intendance AFP
    \",\n          \"Employ\u00e9 en intendance AFP\",\n          \"Employ\u00e9 en
    restauration AFP\",\n          \"Enqu\u00eateur de douane avec dipl\u00f4me f\u00e9d
    \u00e9ral\",\n          \"Entra\u00eeneur de sport de performance avec brevet f\
    u00e9d\u00e9ra...
```

---

**Sample 2: True Complexity: 3.5, Pred Complexity: 4.26, K value: 137, Activated Features: 137**

```
max-age=31536000; includeSubDomains]\n       x-aspnet-version: [4.0.30319]\n        x-
    content-type-options: [nosniff]\n      x-ms-ratelimit-remaining-subscription-
    resource-requests: ['11998']\n      x-powered-by: [ASP.NET]\n      status: {code:
    201, message: Created}\n- request:\n      body: null\n      headers:\n        Accept: [
    application/json]\n      Accept-Encoding: ['gzip, deflate']\n        CommandName: [
    network dns zone import]\n        Connection: [keep-alive]\n        Content-Type: [
    application/json; charset=utf-8]...
```

---

**Sample 3: True Complexity: 5.2, Pred Complexity: 5.39, K value: 187, Activated Features: 187**

```
very much interested in the idea of sanctuary. How do the spirits of these two authors
     and the respective sanctuaries they sought infuse Tom and Nathan\u2019s
    interactions? What other giants of American literature have an influence, direct
    or indirect, on the characters in The Brooklyn Follies?\n\n\u201cYou love life,\
    u201d says Nathan to Tom, \u201cbut you don\u2019t believe in it. And neither do
    I.\u201d This statement quickly becomes untrue as both men cast off their inertia
    . To what extent does action create belief for bo...
```

---

**Sample 4: True Complexity: 6.2, Pred Complexity: 6.05, K value: 216, Activated Features: 216**

```
itself implement the\n    // interface because that exposes all the public methods of
    that interface at the manager level.\n    private static final String
    INTENT_URL_PREFIX = \"intent:\";\n\n    // The animation duration of a URL being
    promoted to a tab when triggered by an\n    // intercept navigation. This is
    faster than the standard tab promotion animation\n    // so that it completes
    before the navigation.\n    private static final long
    INTERCEPT_NAVIGATION_PROMOTION_ANIMATION_DURATION_MS = 40;\n\n  ...
```

---

**Sample 5: True Complexity: 9.1, Pred Complexity: 9.19, K value: 298, Activated Features: 242**

```
parameters of the MSSM and to trace back, sector-wise, the sensitivity to initial
    conditions of the Yukawa couplings and the soft susy breaking parameters. We have
     established analytically a generic screening of non-universality, in the
    vicinity of the infrared quasi fixed points. In practice, this property gives the
     general trend of the behaviour, despite the large number of free parameters, and
     even when one is not very close to such a quasi fixed point. This shows that non
    -universality of th...
```

---

**Sample 6: True Complexity: 9.6, Pred Complexity: 8.91, K value: 294, Activated Features: 294**

```
{{\\ensuremath{\\mathbbm{R}}}}^d) \\to [0,\\infty)$ satisfy for all $d\\in {{\\
    ensuremath{\\mathbbm{N}}}}$, $x=(x_1,\\ldots,x_d)\\in {{\\ensuremath{\\mathbbm{R
    }}}}^d$ that $\n{\\mathbf{A}_{d}}(x)= \\left(\\max\\{x_1,0\\},\\ldots,\\max\\{x_d
    ,0\\}\\right)\n$ and $\\|x\\|=[\\sum_{i=1}^d(x_i)^2]^{1/2}$, let $\n{\\mathbf{N
    }}= \\cup_{H\\in  {{\\ensuremath{\\mathbbm{N}}}}}\\cup_{(k_0,k_1,\\ldots,k_{H+1})
    \\in {{\\ensuremath{\\mathbbm{N}}}}^{H+2}}\n[ \\prod_{n=1}^{H+1} \\left({{\\
    ensuremath{\\mathbbm{R}}}}^{k_{n}}\\times k_{n-1}} \\times{{\\ensurem...
```

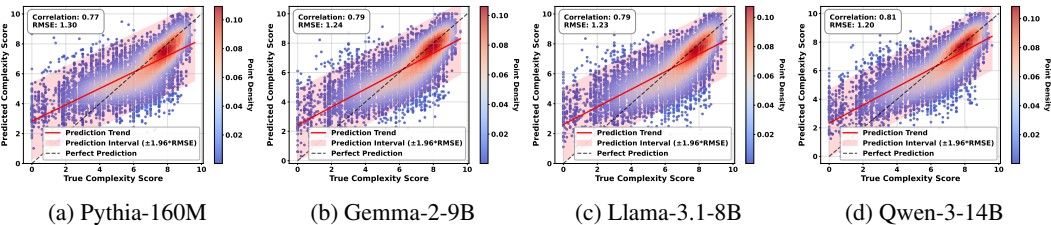

| (a) Pythia-160M | (b) Gemma-2-9B | (c) Llama-3.1-8B | (d) Qwen-3-14B |

Figure 10: Supplement to the results of Figure 3

## C ADDITIONAL SAE TRAINING DETAILS

Our SAEs are trained on the residual stream because researchers typically focus on this component when interpreting or steering model behaviors. This alignment ensures the learned representations directly support the most common SAE applications in model analysis and intervention.

The training and testing datasets for the AdaptiveK SAE are consistent with those utilized in the linear probe training phase. The overall training process is determined by the current step value, with three distinct phases: step=0 is the pre-training phase, dedicated to probe training; step $<$ total steps$\times$phase ratio involve training the SAE while maintaining frozen probe parameters; step $>$ this threshold initiate the joint fine-tuning phase. The total step count is calculated by dividing the total token count (250,000) by the batch processing capacity (2048 tokens), with phase ratio set at 0.9.

During the third phase, the deviation weight adapts dynamically throughout training. The system maintains records of probe losses from the three most recent steps and calculates the rate of loss change. When rapid loss reduction occurs (change rate exceeding the threshold 0.5), the deviation weight is reduced to 0.8 of its original value; otherwise, it increases to 1.2, with an upper limit of 0.5. A sigmoid function maps predicted complexity scores (0-10) to feature quantity ranges (from min_k to max_k, established at 20 to 320), with the sigmoid midpoint corresponding to base_k (80) and steepness (0.6) controlling the mapping curve's configuration. This enables the AdaptiveK SAE to dynamically allocate sparsity by assigning fewer features to simpler texts (low complexity) while allocating more features to complex texts (high complexity).

As the AdaptiveK SAE was trained on 250,000 token activations, we employed identical training and testing datasets for training and evaluating baseline SAEs, with their sparsity configurations detailed in Table 2.

Table 2: Sparsity setting of baseline SAEs

| SAE | Sparsity | Pythia | Gemma/Llama/Qwen/Phi |
|---|---|---|---|
| TopK | | | 20, 40, 80, 160, 320, 640 |
| Batch TopK | K Value | | 20, 40, 80, 160, 320, 640 |
| Matryoshka | | | 20, 40, 80, 160, 320, 640 |
| Gated | | 0.6, 0.9, 1.2, 2, 3, 4 | 0.012, 0.015, 0.02, 0.03, 0.04, 0.06 |
| Relu | Sparsity Penalities | 0.6, 0.9, 1.2, 2, 3, 4 | 0.012, 0.015, 0.02, 0.03, 0.04, 0.06 |
| Relu_new | | 0.6, 0.9, 1.2, 2, 3, 4 | 0.012, 0.015, 0.02, 0.03, 0.04, 0.06 |
| P Anneal | | 0.3, 0.45, 0.6, 1, 1.5, 2 | 0.006, 0.008, 0.01, 0.015, 0.02, 0.025 |

## D MORE RESULTS

### D.1 LINEAR PROBE PERFORMANCE

Figure 10 presents linear probe performance across a broader set of LLMs.

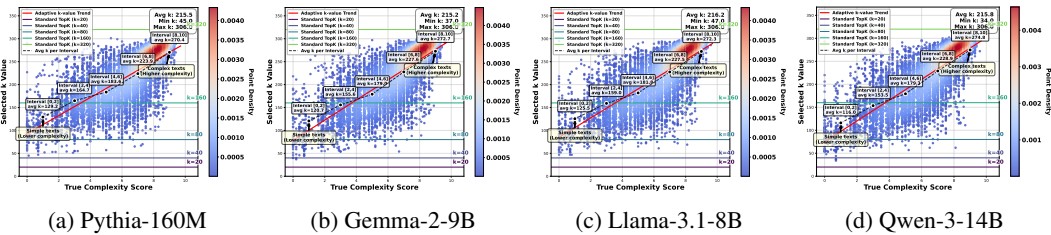

| (a) Pythia-160M | (b) Gemma-2-9B | (c) Llama-3.1-8B | (d) Qwen-3-14B |

Figure 11: Supplement to the results of Figure 4

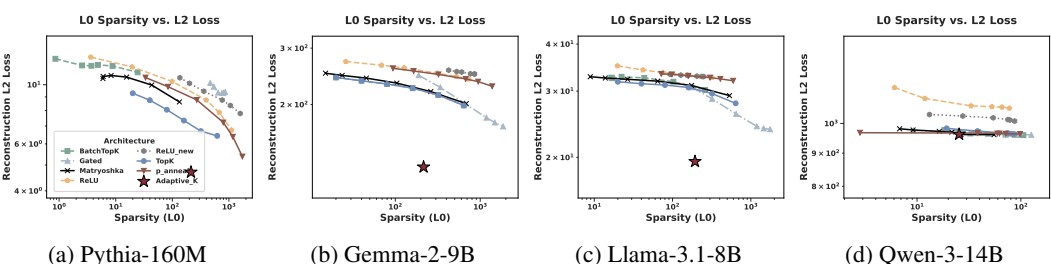

| (a) Pythia-160M | (b) Gemma-2-9B | (c) Llama-3.1-8B | (d) Qwen-3-14B |

Figure 12: Supplement to the results of Figure 5

## D.2 RELATIONSHIP BETWEEN COMPLEXITY SCORES AND ALLOCATED FEATURE COUNTS

Figure 11 presents the relationship between complexity scores and allocated feature counts across a broader set of LLMs.

## D.3 PARETO FRONTIER RESULTS

Figure 12, 13 and 14 presents L2 Loss, Unexplained Variance and Cosine Similarity pareto frontier results across a broader set of LLMs.

# E ADDITIONAL LINEAR PROBE EVALUATION

## E.1 PCA DIMENSIONALITY REDUCTION EXPERIMENTS

To provide more rigorous statistical validation, PCA dimensionality reduction experiments were conducted on the activation data. Specifically, since the original probe dimensionality is 2048 (for Gemma-2-2B), it could theoretically "memorize" substantial information. However, if complexity is truly linearly encoded, then only a few dimensions should be needed for accurate prediction.

We performed PCA decomposition on all context activation vectors, identifying the top k directions with maximum variance (principal components), and projected the original 2048-dimensional activations onto these k directions (k=10-500). Probes for complexity prediction were then trained on the low-dimensional representations. Figure 15 shows the RMSE and Pearson correlation coefficients of probes using different numbers of principal components. Using just 200 principal components (9.8% of full dimensionality), the probe achieves a Pearson correlation of 0.72 compared to 0.79 for the full-dimensional probe, recovering 91% of the predictive performance while explaining 75.6% of the variance. With 400 components (19.5% of dimensions), performance reaches 96% of the full probe (Pearson: 0.76 vs 0.79) with 88.4% explained variance. This demonstrates that text complexity is indeed linearly encoded. If it were merely high-dimensional memorization, many more principal components would be required to achieve good predictive performance.

## E.2 LAYER-WISE EVALUATION

To further validate the effectiveness of complexity prediction using linear probes, we conducted layer-wise experiments across all 26 layers of Gemma-2-2B to understand how complexity repre-

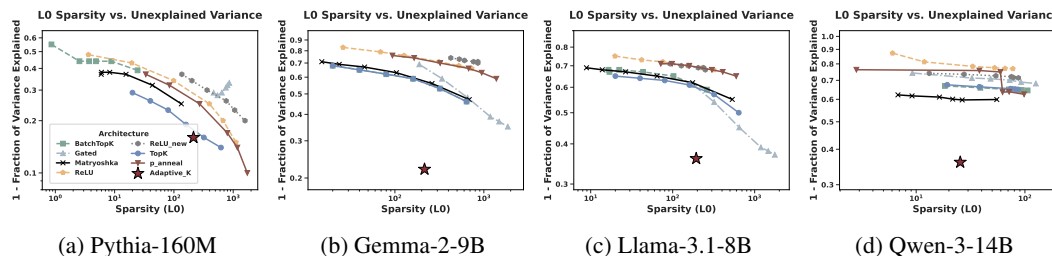

(a) Pythia-160M  (b) Gemma-2-9B  (c) Llama-3.1-8B  (d) Qwen-3-14B

Figure 13: Supplement to the results of Figure 6

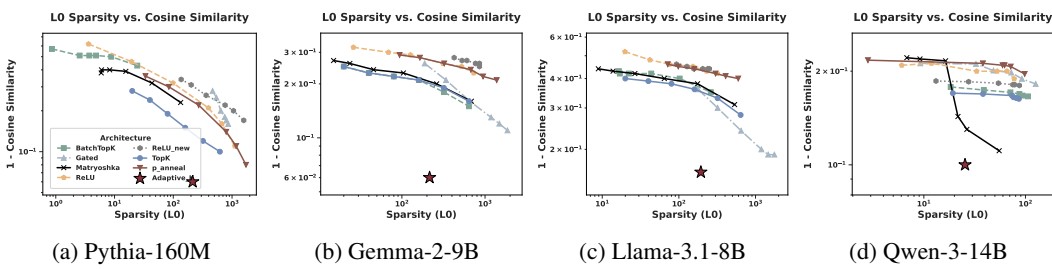

(a) Pythia-160M  (b) Gemma-2-9B  (c) Llama-3.1-8B  (d) Qwen-3-14B

Figure 14: Supplement to the results of Figure 7

sentations develop throughout the model. Following previous work (Gurnee & Tegmark, 2023), which found that probing performance exhibits a characteristic pattern of initial growth followed by saturation as layer depth increases, our experimental results in Figure 16 demonstrate the same trajectory. Starting from layer 4 with a Pearson correlation of 0.766, performance steadily improves through the middle layers, reaching peak performance at layer 22 with a correlation of 0.814 and RMSE of 1.18. Beyond this point, performance plateaus and even slightly decreases (layer 24: 0.801). This demonstrates that LLM representations contain complexity information in the same way they contain spatial and temporal information, and that deeper layers are progressively better at capturing text complexity.

## F  ADDITIONAL SAE EVALUATION

### F.1  LAYER-WISE EVALUATION

We trained AdaptiveK SAE on each layer of Pythia-160M and on layers 4, 8, 12, 16, 20, and 24 of the Gemma-2-2B model. A cross-layer comparison of key performance metrics is presented in Figure 17 and 18. L2 ratio measures the proportion between the L2 norms of reconstructed and original activations, with values closer to 1 indicating preservation of the original activation magnitude. AdaptiveK exhibits robust performance across all tested layers in both models, with Explained Variance, Cosine Similarity, and L2 Ratio consistently above 0.74, 0.91, and 0.89 respectively. This confirms the algorithm's generalizability throughout the entire LLM hierarchy.

### F.2  EXTENDING TO ENCODER-ONLY AND ENCODER-DECODER MODELS

We extended our model to encoder-only BERT and encoder-decoder T5, with results shown in Table 3. BERT-340M achieved 0.89 cosine similarity and 0.82 L2 ratio, demonstrating successful adaptation to bidirectional attention mechanisms. T5-small results show the encoder outperforms the decoder by 23.56% in explained variance, indicating that input understanding tasks are relatively regular while generation tasks are more complex. The encoder also performs better on other metrics, suggesting that encoders focus on understanding input semantic representations while decoders must handle both understanding and generation tasks simultaneously. All results maintain good reconstruction quality (cosine similarity more than 0.89, L2 ratio close to 1.0), proving the cross-architecture universality of "complexity → more features".

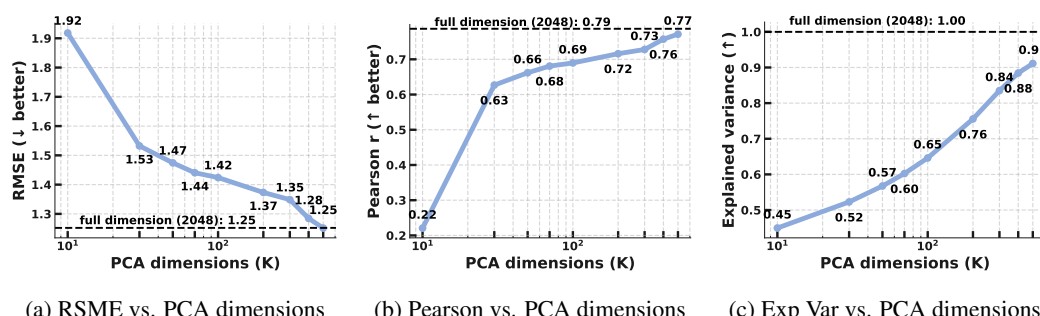

(a) RSME vs. PCA dimensions  (b) Pearson vs. PCA dimensions  (c) Exp Var vs. PCA dimensions

Figure 15: PCA dimensionality reduction experiments on Gemma-2-2B

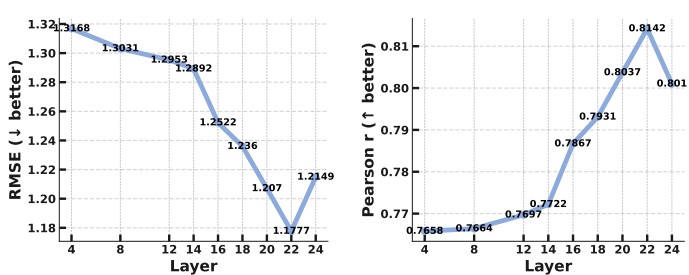

Figure 16: Layer-wise evaluation of linear probe on Gemma-2-2B

### F.3 OTHER EVALUATION METRICS

In this section, we evaluate the AdaptiveK SAE using five metrics from SAEBench (Karvonen et al., 2025). For the Feature Absorption metric, we directly compare the results of AdaptiveK SAE with those of the baseline SAEs reported in SAEBench, **noting that their training dataset was 2000 times larger than ours**. For the remaining metrics, (Spurious Correlation Removal, Targeted Probe Perturbation, Resolving Attribute-Value Entanglements in Language Models, and Sparse Probing), the AdaptiveK SAE is compared against baseline SAEs trained on an identical amount of training data.

#### F.3.1 FEATURE ABSORPTION

One of the primary objectives of SAEs is to enhance feature interpretability through sparse activation patterns. However, when concepts exhibit hierarchical relationships, concept A (*e.g.*, red) inherently implies a broader concept B (*e.g.*, color), instead of dedicating separate, clear latents for both A and B, SAEs tend to develop a latent unit representing A and another representing "B except A". While this approach optimizes sparsity, it significantly compromises interpretability.

Following (Karvonen et al., 2025), Feature Absorption is measured using a first-letter classification task. It first establishes a "ground truth" directional vector $p$, for each first-letter concept by training linear probes on the base language model's activations ($a_{model}$). It then identifies a set of "main"

Table 3: Performance of encoder-only and encoder-decoder models

| Model | Layer | SAE Position | Explained Variance | Cosine Similarity | L2 Ratio |
|---|---|---|---|---|---|
| BERT-340M | 8 | encoder | 0.66 | 0.89 | 0.8225 |
| T5-small | 3 | encoder | 0.97 | 0.98 | 1.0093 |
| T5-small | 3 | decoder | 0.74 | 0.97 | 1.0139 |

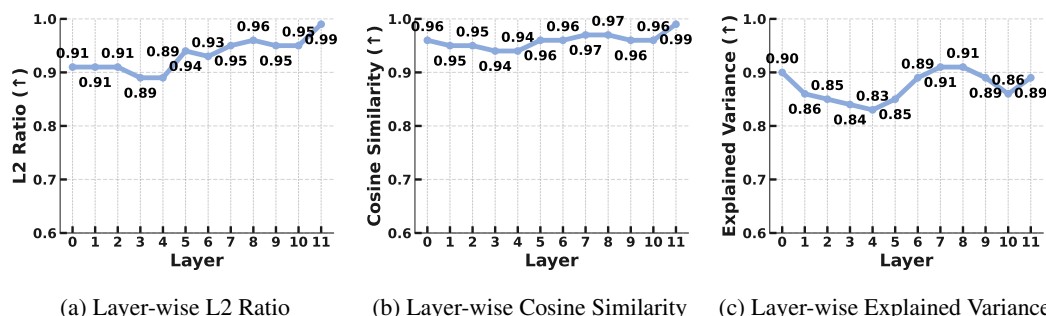

(a) Layer-wise L2 Ratio  (b) Layer-wise Cosine Similarity  (c) Layer-wise Explained Variance

Figure 17: Layer-wise performance on Pythia-160M

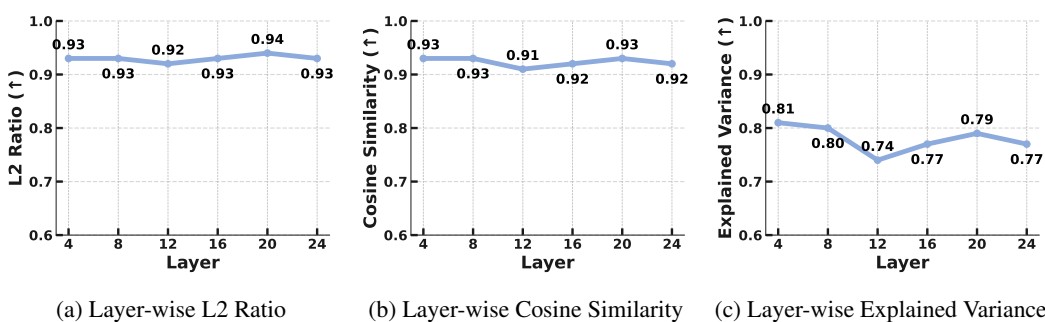

(a) Layer-wise L2 Ratio  (b) Layer-wise Cosine Similarity  (c) Layer-wise Explained Variance

Figure 18: Layer-wise performance on Gemma-2-2B

SAE latents ($S_{main}$) expected to represent each letter. The core of the measurement involves analyzing individual instances (words). For an instance, if the main latents don't fully capture the ground truth signal (i.e., $\sum_{i \in S_{main}} a_i d_i \cdot p < a_{model} \cdot p$, where $a_i$ a is latent activation and $d_i$ its decoder vector), and other "absorbing" latents ($S_{abs}$) that align with $p$ compensate for this deficit, an absorption fraction is calculated. Let $P_{main} = \sum_{j \in S_{main}} a_j d_j \cdot p$ be the projection from main features, and $P_{compensated\_by\_absorbers}$ be projection from the top few absorbing latents that cover the signal portion not captured by $P_{main}$. The instance-level absorption fraction $f_{abs}$ is then:

$$f_{\text{abs}} = \frac{P_{\text{compensated\_by\_absorbers}}}{P_{\text{compensated\_by\_absorbers}} + P_{\text{main}}}. \tag{13}$$

This value closer to 0 means less feature absorption. We calculate two metrics: Mean Absorption Fraction per letter is the average of these $f_{abs}$ values over all relevant instances for that letter. Separately, an instance is marked for full absorption if stricter binary criteria are met: essentially, if main features are inactive and a single, dominant non-main latent (aligned with $p$) overwhelmingly represents the letter's ground truth signal. The Full Absorption Rate per letter is simply the proportion of relevant instances for that letter which meet these conditions for "full absorption", indicating how often extreme absorption occurs.

Utilizing this metric, we assessed the performance of our AdaptiveK SAE on layer 3 of Pythia-160M and layer 12 of Gemma-2-2B, as shown in Figure 19 and 20. Notably, when directly benchmarked against SAEs from SAEBench (Karvonen et al., 2025) for Gemma-2-2B's layer 12, AdaptiveK exhibited superior results on both metrics (Figure 21). This outperformance is particularly significant given that our AdaptiveK was trained on only 250,000 tokens, a dataset 2000 times smaller than the 500,000,000 tokens used for the SAEBench models. In AdaptiveK, concepts are correctly allocated to their intended primary latents rather than being dispersed across unrelated variables. Cases where concepts are entirely misrepresented in non-primary latents are notably rare. This demonstrates AdaptiveK's exceptional effectiveness in maintaining conceptual integrity and preventing feature fragmentation.

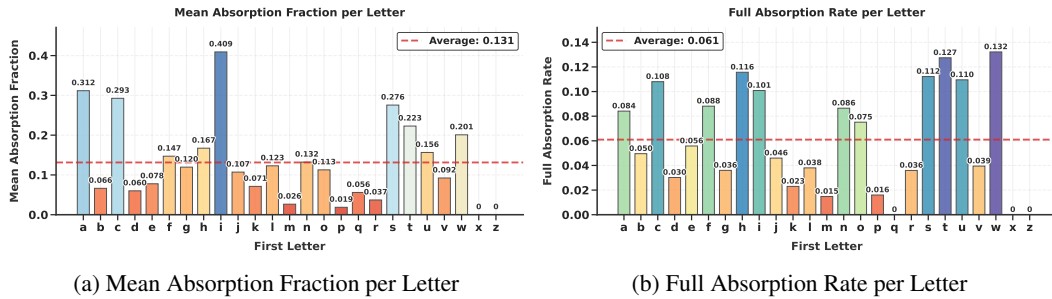

(a) Mean Absorption Fraction per Letter

(b) Full Absorption Rate per Letter

Figure 19: Letter Absorption Results on Pythia-160M

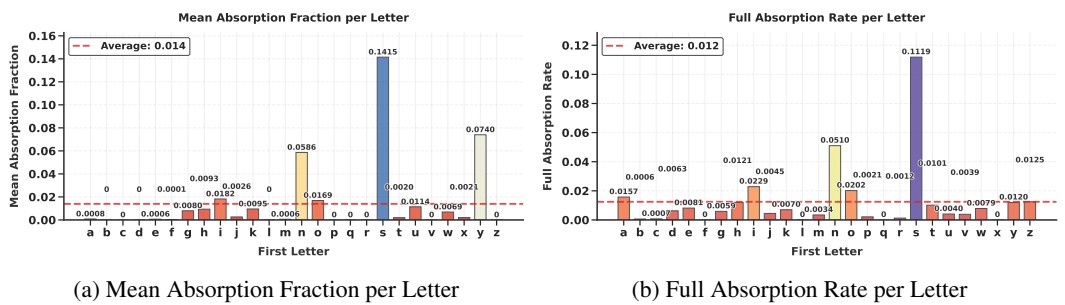

(a) Mean Absorption Fraction per Letter

(b) Full Absorption Rate per Letter

Figure 20: Letter Absorption Results on Gemma-2-2B

### F.3.2 SPURIOUS CORRELATION REMOVAL (SCR)

Spurious Correlation Removal evaluates an SAE's ability to disentangle distinct concepts by measuring how effectively it can remove spurious correlations from classifiers. Likewise, utilizing method from SAEBench (Karvonen et al., 2025), we first generated biased datasets containing spurious correlations (*e.g.*, professor+male and nurse+female from the Bias in Bios dataset). A linear classifier is then trained on this biased dataset, learning to rely on both the target concept (profession) and the spurious concept (gender). To evaluate an SAE, the method identifies latents most strongly associated with the spurious concept (gender) through probe attribution scores. These identified latents are then zero-ablated, creating a modified classifier. The final SCR score is normalized as:

$$\text{SCR Score} = \frac{A_{\text{abl}} - A_{\text{base}}}{A_{\text{oracle}} - A_{\text{base}}}, \tag{14}$$

where $A_{\text{abl}}$ is accuracy after ablation, $A_{\text{base}}$ is baseline accuracy, and $A_{\text{oracle}}$ is the accuracy of a classifier trained directly on the desired concept. Higher SCR scores indicate better concept disentanglement, suggesting the SAE effectively isolates distinct concepts into separate latents.

The results of comparing the AdaptiveK SAE trained on Pythia-160M layer 3 against other baseline SAEs, where all SAEs were trained using 250,000 tokens, are depicted in Figure 22. SCR Top10 and SCR Top20 refer to the SCR scores when ablating the top 10 and top 20 latents respectively that are most associated with the spurious concept. AdaptiveK shows dramatically higher SCR scores in both settings, directly indicating its superior concept disentanglement, with clearer separation between concepts like gender and profession. Additionally, it produces latent representations where concepts are more cleanly isolated in specific latents, enabling more effective debiasing of classifiers.

### F.3.3 TARGETED PROBE PERTURBATION (TPP)

Unlike SCR which works with binary correlated labels, TPP extends SCR to multiclass settings. For each class $i$ in a dataset with $m$ classes, TPP identifies the most relevant latents $L_i$ for that class and trains linear classifiers $C_j$ for each class $j$ with accuracy $A_j$. Then, it creates modified classifiers $C_{i,j}$ by ablating latents $L_i$, with accuracy $A_{i,j}$. We can calculate TPP Score as:

$$\text{TPP Score} = \text{mean}_{i=j}(A_{i,j} - A_j) - \text{mean}_{i \neq j}(A_{i,j} - A_j). \tag{15}$$

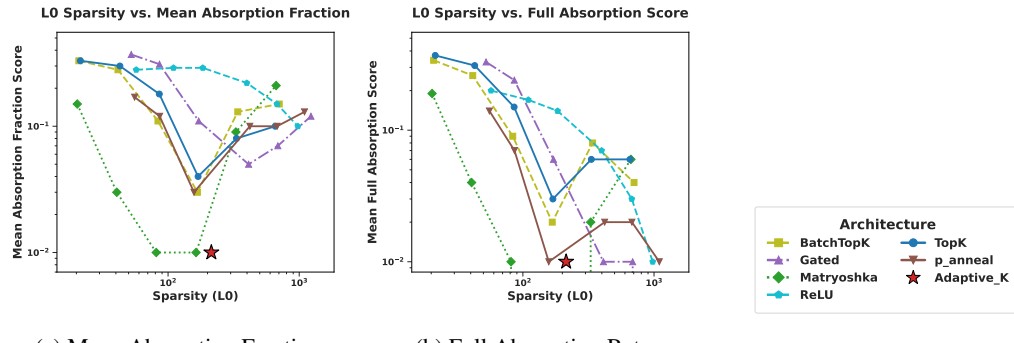

(a) Mean Absorption Fraction          (b) Full Absorption Rate

Figure 21: Two Feature Absorption metrics across different SAEs on Gemma-2-2B Layer 12

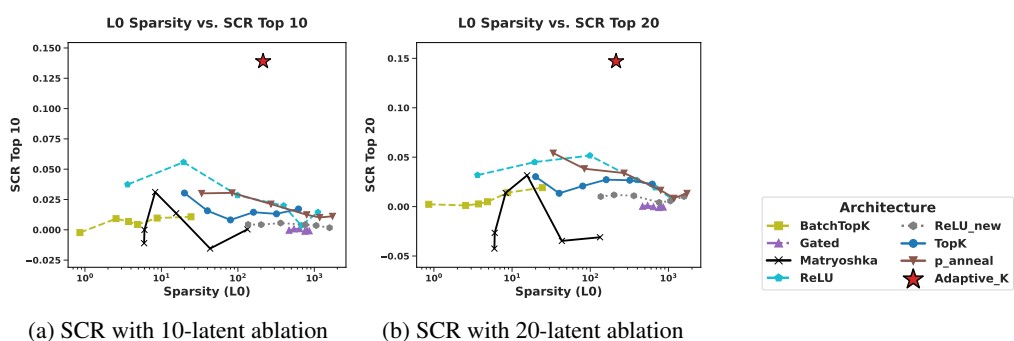

(a) SCR with 10-latent ablation          (b) SCR with 20-latent ablation

Figure 22: SCR scores across two intervention settings on Pythia-160M Layer 3

This formula captures the difference between within-class effects (when $i = j$) and cross-class effects (when $i \neq j$). A high TPP score indicates good disentanglement, ablating latents for class $i$ primarily affects only class $i$'s accuracy while leaving other class accuracies unchanged. This shows concepts are encoded in separate, non-overlapping latent dimensions. In the same way, TPP Top10 and TPP Top20 refer to the TPP scores when ablating the top 10 and top 20 most relevant latents for each class, respectively.

As shown in Figure 23, AdaptiveK outperforms most SAEs, showing that ablating latents identified for one class primarily affects only that class's probe accuracy. This precise targeting indicates AdaptiveK organizes its latent space with clearer conceptual boundaries. Combined with its SCR results in Figure 22, AdaptiveK creates a more structurally organized latent space with minimal concept overlap. While Matryoshka Batch TopK performs well on TPP, AdaptiveK's consistent high performance across both TPP and SCR metrics demonstrates its representation efficiently separates both binary concepts and multiclass classes. This dual strength in disentanglement makes it particularly suited for interpretability tasks requiring precise concept isolation and targeted.

### F.3.4 Resolving Attribute-Value Entanglements in Language Models (RAVEL)

RAVEL directly measures a key application of interpretability, which is the practical utility of an SAE for targeted knowledge editing. It tests whether interventions on specific latents can modify one attribute of an entity while preserving other attributes. The evaluation focuses on whether an SAE can help a language model make targeted factual modifications, for example, changing Paris's country from France to Japan while correctly maintaining that the language spoken there is still French (rather than incorrectly switching to Japanese). The evaluation begins by collecting high-confidence entity-attribute predictions across five diverse categories (cities, Nobel laureates, physical objects, etc.). For each entity, RAVEL identifies which latents most strongly encode specific attributes using trained probes. It then tests what happens when these latents are manipulated - can the model be made to believe Paris is in Japan while still knowing French is spoken there? This capability is measured through two complementary metrics: the cause score (how effectively the intervention

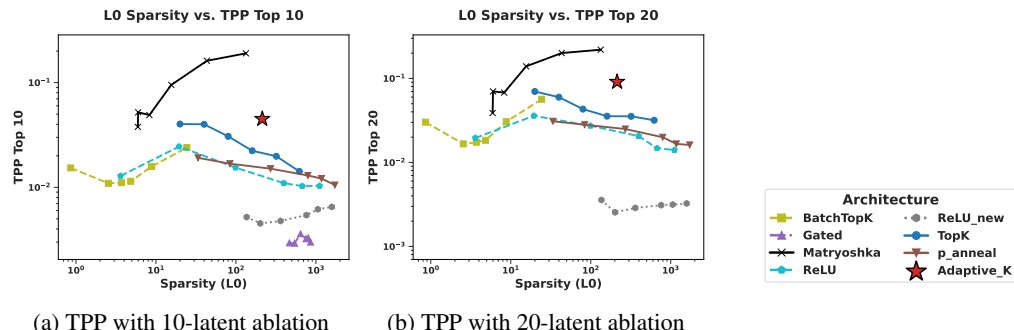

(a) TPP with 10-latent ablation      (b) TPP with 20-latent ablation

Figure 23: TPP scores across two intervention settings on Pythia-160M Layer 3

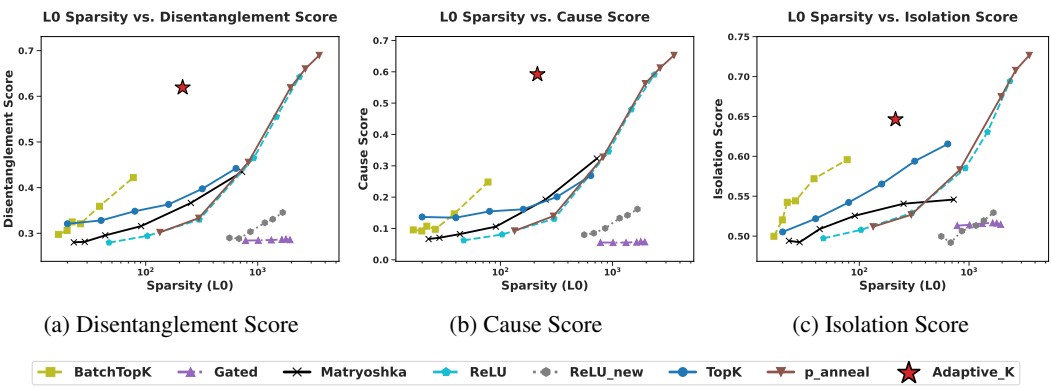

(a) Disentanglement Score    (b) Cause Score    (c) Isolation Score

Figure 24: Three RAVEL results on Gemma-2-2B Layer 12

changes the target attribute) and the isolation score (how well other attributes remain unaffected). These are averaged into a final disentanglement score. Higher scores indicate better attribute separation in the SAE's latent space, showing it has successfully disentangled different factual properties into distinct latent dimensions that can be independently manipulated.

In the disentanglement score (Figure 24a), AdaptiveK achieves 0.62, substantially higher than contemporary architectures like TopK and Matryoshka at comparable sparsity levels. For the cause score (Figure 24b), AdaptiveK reaches 0.6, roughly double the effectiveness of other SAEs at similar sparsity. This indicates AdaptiveK is exceptionally good at identifying and modifying the specific latents that control target attributes (like a city's country). The isolation score (Figure 24c) shows AdaptiveK at 0.65, demonstrating it maintains unrelated attributes more effectively than others. While P Anneal and some other SAEs eventually reach similar or higher disentanglement scores, they require much higher sparsity levels (L0 > 1000) to do so, making them less practical for interpretability work that benefits from more compact representations. Combined with the earlier SCR and TPP results, these RAVEL findings confirm that AdaptiveK creates a latent space with exceptionally clean separation between different concepts and attributes, enabling more precise and controlled interventions on language model knowledge.

### F.3.5 SPARSE PROBING

Unlike other metrics that focus on concept separation, Sparse Probing evaluates an SAE's ability to organize meaningful semantic features by measuring how effectively it concentrates concept-specific information in individual latents. The method applies the SAE to encode texts from five diverse datasets covering profession classification (Bias in Bios), product categorization and sentiment analysis (Amazon Reviews), language identification (Europarl), programming language detection (GitHub), and news topic categorization (AG News). For each of the 35 binary classification tasks, the evaluation first identifies which latents show the greatest activation difference between positive and negative examples. A logistic regression probe is then trained using only these selected

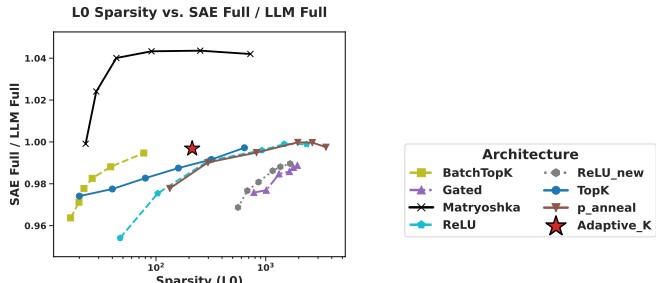

Figure 25: SAE Full Accuracy / LLM Full Accuracy on Gemma-2-2B Layer 12

latents (ranging from just the single most relevant latent to the top 5), with performance measured on 1,000 held-out test examples. The key insight is that if the SAE has effectively organized information, even a small number of latents (the top-K) should contain sufficient information to perform specific classification tasks.

Metric "Full Activations Accuracy" represents classification performance when using all SAE activations, establishing an upper bound. Metrics "Top-K Accuracy" (where K is 1, 2, or 5) measure performance when restricting the probe to only the K most relevant latents for each task. First, we measure information retention rate (SAE Full Activations Accuracy / LLM Full Activations Accuracy), which quantifies how well each SAE preserves the original model's information when using all reconstructed activations. As shown in Figure 25, AdaptiveK's retention ratio of approximately 0.997 demonstrates near-perfect preservation of the original model's information. While Matryoshka's slightly higher ratio ($>$1) suggests beneficial feature reorganization or denoising during reconstruction, such enhancement represents a supplementary advantage rather than a necessity.

Second, we examine relative feature concentration (SAE Top-K / LLM Top-K) across three granularities (K=1,2,5) as illustrated in Figure 26. These metrics reveal how efficiently each architecture concentrates concept-specific information in its most relevant latents compared to the base model. The SAE Top-K/LLM Top-K ratios for AdaptiveK consistently exceed 1. Though marginally below Matryoshka, these values convincingly demonstrate AdaptiveK's superior ability to concentrate concept-relevant information in fewer latent dimensions than the original model requires, indicating more efficient latent space organization.

Finally, we assess information concentration efficiency (SAE Top-K / SAE Full) in Figure 27, which measures how much of an SAE's total information is captured in its K most relevant latents. AdaptiveK captures approximately 82% of its complete representation using just its most relevant single latent variable, surpassing most other SAE architectures and demonstrating exceptional information compression.

Collectively, these metrics establish AdaptiveK's balanced excellence: it maintains original model information (retention rate), concentrates more concept-relevant information in fewer dimensions than the original model (relative feature concentration), and achieves a highly organized internal representation that localizes most information in a minimal number of latents (information concentration efficiency).

### F.4 HYPERPARAMETERS ANALYSIS

#### F.4.1 REGULARIZATION STRENGTH $\lambda$

As stated in Section 3.1.1, to determine the optimal regularization strength $\lambda$, we perform 5-fold cross-validation. For each $\lambda$ in the set $\{0.001, 0.01, 0.1, 1.0, 10.0, 100.0, 1000.0\}$, the probe is trained on four folds and evaluated on the remaining fold using root mean squared error (RMSE). Table 4 reports the average RMSE across folds for each $\lambda$ on selected models, with the best $\lambda$ highlighted in bold.

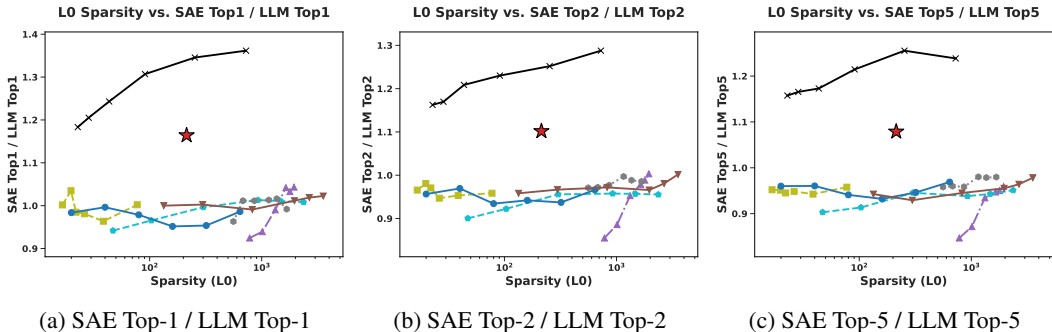

(a) SAE Top-1 / LLM Top-1     (b) SAE Top-2 / LLM Top-2     (c) SAE Top-5 / LLM Top-5

Figure 26: SAE Top-K / LLM Top-K on Gemma-2-2B Layer 12

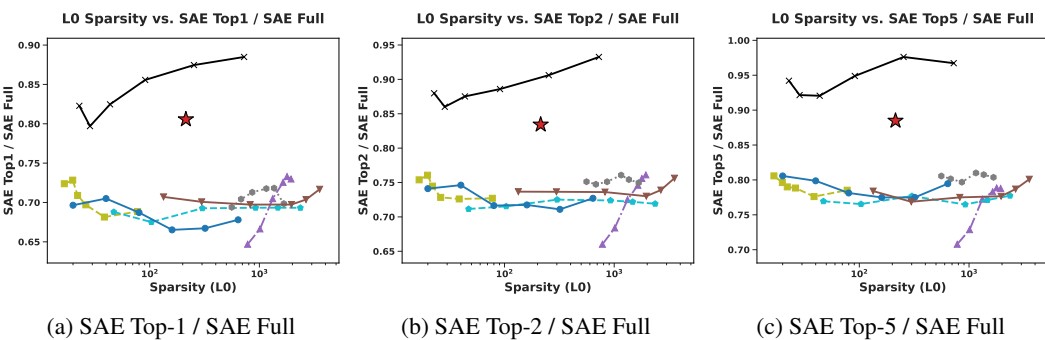

(a) SAE Top-1 / SAE Full     (b) SAE Top-2 / SAE Full     (c) SAE Top-5 / SAE Full

Figure 27: SAE Top-K / SAE Full on Gemma-2-2B Layer 12

### F.4.2 STEEPNESS OF SIGMOID FUNCTION $s$

$s$ determines how complexity scores map to k-values. Our experiments on Gemma-2-2B (Table 5) show that $s$ exhibits remarkable robustness across the tested range of 2.0 to 12.0. As $s$ increases, the k-value distribution becomes more dynamic: lower values ($s$=2.0) produce conservative allocation with a narrow range (min k=143, max k=225), while higher values ($s$=12.0) enable more aggressive feature allocation with expanded ranges (min k=58, max k=315). Crucially, reconstruction quality remains stable across all tested values. These results indicate that our method is highly robust to $s$ parameter selection, with performance variations of less than 1% across the entire range, validating our choice of $s$=6.0 as a balanced default configuration.

Table 4: Performance across models with varying $\lambda$ values

| $\lambda$ | Gemma-2-2b | Gemma-2-9b | Llama-3.1-8b | Qwen-3-8b | Qwen-3-14b | Phi-4-14b |
|---|---|---|---|---|---|---|
| 0.001 | 1.1937 | 1.2663 | 1.2612 | 1.0875 | 1.2425 | 1.1934 |
| 0.01 | 1.1937 | 1.2663 | 1.2612 | 1.0875 | **1.2424** | 1.1934 |
| 0.1 | 1.1937 | 1.2663 | 1.2612 | 1.0875 | 1.2425 | 1.1934 |
| 1.0 | 1.1937 | **1.2662** | 1.2612 | 1.0875 | 1.2425 | 1.1934 |
| 10.0 | 1.1937 | 1.2663 | 1.2611 | 1.0874 | 1.2424 | 1.1934 |
| 100.0 | **1.1935** | 1.2663 | **1.2605** | **1.0873** | 1.2424 | **1.1933** |
| 1000.0 | 1.1937 | 1.2663 | 1.2608 | 1.0875 | 1.2424 | 1.1933 |

Table 5: Effect of $s$ on $k$ statistics and performance metrics

| $k$ on test set | min $k$ | max $k$ | avg $k$ | Explained Variance | Cosine Similarity | L2 Ratio |
|---|---|---|---|---|---|---|
| $s = 2.0$ | 143 | 225 | 188 | 0.738 | 0.908 | 0.911 |
| $s = 4.0$ | 118 | 267 | 203 | 0.741 | 0.909 | 0.914 |
| $s = 6.0$ | **96** | **291** | **214** | **0.743** | **0.909** | **0.921** |
| $s = 8.0$ | 80 | 304 | 222 | 0.743 | 0.909 | 0.916 |
| $s = 12.0$ | 58 | 315 | 232 | 0.742 | 0.909 | 0.917 |

### F.4.3 PROBE WEIGHT $\gamma$

$\gamma$ balances the SAE reconstruction loss and probe loss during joint fine-tuning, where smaller values prioritize reconstruction quality and produce more conservative k-value allocation, while larger $\gamma$ values enhance probe dominance and expand the k-value range to better reflect complexity differences. Our experiments (Table 6) confirm this expected behavior, with k-value ranges expanding as $\gamma$ increases (from min k=98, max k=285 at $\gamma$=0.5 to min k=92, max k=293 at $\gamma$=1.0), but across $\gamma \in [0.5, 1.0]$ show minimal performance variation. This stability demonstrates that our method is robust to $\gamma$ selection, with performance variations under 1%, validating our default choice of $\gamma$=0.9.

Table 6: Effect of $\gamma$ on $k$ statistics and performance metrics

| $k$ on test set | min $k$ | max $k$ | avg $k$ | Explained Variance | Cosine Similarity | L2 Ratio |
|---|---|---|---|---|---|---|
| $\gamma = 0.5$ | 98 | 285 | 208 | 0.742 | 0.909 | 0.913 |
| $\gamma = 0.7$ | 97 | 289 | 210 | 0.743 | 0.909 | 0.916 |
| $\gamma = 0.9$ | **96** | **291** | **214** | **0.743** | **0.909** | **0.921** |
| $\gamma = 1.0$ | 92 | 293 | 215 | 0.743 | 0.909 | 0.919 |

### F.4.4 DEVIATION PENALTY $\delta$

$\delta$ prevents probe parameters from deviating too far from their pre-trained values. Starting with $\delta$=0.2, we dynamically adjust this weight during training by monitoring probe loss changes over the recent 3 steps. We calculate the loss change rate as $\frac{\text{earliest loss} - \text{latest loss}}{\text{earliest loss}}$ and adjust $\delta$ accordingly: when loss decreases rapidly (change rate $> 0.05$), indicating good probe learning progress in complexity prediction, we reduce the deviation constraint by setting $\delta = \text{current}_\delta \times 0.8$ (minimum 0.01) to allow more parameter flexibility; when loss stagnates or increases, suggesting learning difficulties or overfitting, we strengthen the constraint by setting $\delta = \text{current}_\delta \times 1.2$ (maximum 0.5) to prevent excessive parameter drift. The upper bound of 0.5 is critical because at this value, the deviation loss becomes comparable in magnitude to the probe loss, and higher values would make the deviation penalty too dominant, completely preventing the probe from adapting to new patterns when learning becomes difficult. Additionally, since deviation loss measures distance from initial values, excessive weights would create large gradients that harm training stability.

### F.4.5 REGARDING SIGMOID-BASED TRANSFORMATION

The sigmoid function provides a smooth non-linear mapping that aligns well with empirical observations about complexity–feature relationships. Our implementation uses only two intuitive parameters: mid_point, which specifies the complexity level corresponding to the base $k$, and steepness, which controls the transition smoothness. In practice, we found these parameters to be stable across models. Setting mid_point=0.5 centers the allocation around normalized complexity, while steepness=6.0 ensures appropriate transition smoothness. To test robustness, we also experimented with Linear Mapping and Exponential Mapping, with results on Gemma-2-2B reported in Table 7. All three mapping methods achieved similar performance across metrics, demonstrating that our approach is not sensitive to the specific choice of mapping.

Table 7: Comparison of mapping methods

| Method | avg $k$ | min $k$ | max $k$ | Explained Var | Cosine Sim | L2 Ratio | Rel Recon Bias |
|---|---|---|---|---|---|---|---|
| Sigmoid | 214 | 95 | 291 | 0.80 | 0.93 | 0.93 | 0.9995 |
| Linear | 206 | 98 | 296 | 0.79 | 0.93 | 0.92 | 0.9988 |
| Exponential | 144 | 53 | 271 | 0.78 | 0.92 | 0.92 | 0.9949 |

### F.5 SAE PERFORMANCE WITH LARGER K VALUE

Additional experiments have been conducted with different $k_{min}$ and $k_{max}$. As shown in Table 8, when scaling $k_{max}$ from 320 to 640, we observe steady improvements: Explained Variance increases from 0.743 to 0.789, Cosine Similarity from 0.909 to 0.926, and L2 Ratio from 0.921 to 0.935. These results indicate that AdaptiveK benefits from increased capacity similarly to standard SAEs.

Table 8: Performance with different $k_{\min}$ and $k_{\max}$ settings

| $k$ on test set | min $k$ | max $k$ | avg $k$ | Explained Variance | Cosine Similarity | L2 Ratio |
|---|---|---|---|---|---|---|
| $k_{\min} = 20$, $k_{\max} = 320$ | 96 | 291 | 214 | 0.743 | 0.909 | 0.921 |
| $k_{\min} = 20$, $k_{\max} = 480$ | 132 | 435 | 313 | 0.768 | 0.919 | 0.926 |
| $k_{\min} = 20$, $k_{\max} = 640$ | 170 | 579 | 415 | 0.789 | 0.926 | 0.935 |

## G TRAINING EFFICIENCY ANALYSIS

In large-scale or token-level training scenarios, complexity annotation is often regarded as a scalability challenge, since exhaustive annotation may become costly. AdaptiveK addresses this issue through several design considerations.

Firstly, annotation is a separate process. Complexity annotation occurs during the data preprocessing stage as targets for complexity prediction, independent of SAE's three-stage training. Once completed, the annotations can be used to train linear probes for any model architecture. Secondly, for large-scale training data, only a subset is annotated for linear probe training while using the full dataset for SAE training.

Training efficiency is further assessed through comparisons across multiple SAE configurations. While traditional SAEs avoid complexity annotation, they require training multiple SAEs with different sparsity settings (different k values or sparsity penalties). Table 9 shows complete training times for other SAEs with single sparsity configurations on Gemma-2-2B, all exceeding AdaptiveK's training time. For six different sparsity levels (*e.g.*, k=20,40,80,160,320,640), the total training time would exceed AdaptiveK by more than 6-fold.

Table 9: Total training time (minutes) for different SAEs

| | AdaptiveK | Batch TopK | Gated | Matryoshka | P Anneal | Relu | Relu New | TopK |
|---|---|---|---|---|---|---|---|---|
| Total (min) | 11084 | 13853 | 13908 | 13773 | 13902 | 13835 | 13814 | 13955 |

## H ADAPTABILITY TO TOKEN-LEVEL EVALUATION

Although AdaptiveK operates at the context level by default, its design also enables reliable token-level evaluation. The training process uses contexts of 1024 tokens each (as mentioned in Section 3.1.1). These contexts contain multiple sentences of varying lengths, including truncated incomplete sentence. The complexity of the last token in each context is used to represent the entire context,

ensuring generalization capability. This deliberate approach of not using complete sentences enables training with the last token regardless of sentence length (whether 32 tokens, 256 tokens, etc.).

Due to training process, during evaluation, even when the input context is not 1024 tokens (for example, the first 256 or 500 tokens), our method can still effectively compute complexity based on the representation of the last token in the first n tokens. This is the reason and mechanism for why and how it can effectively predict the complexity of tokens at arbitrary positions.

Empirical results on Gemma-2-9B further confirm this adaptability. As shown in Table 10 reconstruction performance is consistent across token positions 10, 100, 200, 500, 800, and 1000. The cosine similarity varies by only 0.3% (0.954-0.957), indicating highly consistent reconstruction directions. L2 ratios and reconstruction bias remain close to 1.0 (0.9583-0.9701, 1.0052-1.0208), demonstrating accurate reconstruction magnitudes. All metrics vary minimally ($< 5\%$) across positions, proving that AdaptiveK SAE maintains stable reconstruction quality at any context position. Our context-level approach reflects efficiency considerations rather than methodological limitations.

Table 10: Evaluation across different token positions

| Token Position | Cosine Similarity | Explained Variance | L2 Ratio | Recon Bias | K Value |
|:---:|:---:|:---:|:---:|:---:|:---:|
| 10 | 0.9544 | 0.7865 | 0.9701 | 1.0208 | 267 |
| 100 | 0.9570 | 0.8255 | 0.9688 | 1.0130 | 282 |
| 200 | 0.9557 | 0.8268 | 0.9622 | 1.0104 | 285 |
| 500 | 0.9544 | 0.8203 | 0.9674 | 1.0130 | 285 |
| 800 | 0.9557 | 0.8333 | 0.9622 | 1.0104 | 286 |
| 1000 | 0.9557 | 0.8151 | 0.9583 | 1.0052 | 290 |

# I  LIMITATIONS AND BROADER IMPACTS

## I.1  LIMITATIONS

Due to the cost constraints associated with API annotation, our study utilized a significantly smaller training dataset compared to the 500,000,000 tokens employed in SAEBench. Specifically, we trained on 250,000 contexts, extracting the activation value of the final token from each context as its representational vector. Despite this substantial reduction in training data volume, our AdaptiveK SAE achieved impressive performance across most evaluation metrics, with some results even surpassing the baseline SAEs reported in SAEBench. This remarkable efficiency demonstrates the considerable potential of our approach. Rather than relying exclusively on extensive training data, we have introduced a novel SAE training algorithm that fundamentally rethinks sparsity allocation.

## I.2  BROADER IMPACTS

Our AdaptiveK Sparse Autoencoder offers significant broader impacts across multiple domains. By dynamically allocating representational capacity based on input complexity, it enhances computational efficiency through optimized resource utilization, potentially reducing energy consumption in large-scale AI systems. This adaptive approach simultaneously improves model interpretability by establishing clear correlations between complexity metrics and feature activation patterns, providing researchers with new insights into representation learning mechanisms.

---

## Prompt for Scoring Context Complexity

**Detailed Evaluation Dimensions**

1. Lexical Complexity (*Weight: 20%*): Evaluate the vocabulary sophistication level using the following criteria
   - Word Frequency: Proportion of uncommon words (not in the 5000 most frequent words)
   - Word Length: Average syllable count and character length of words
   - Lexical Diversity: Type–token ratio (unique words divided by total words)
   - Technical Terminology: Presence of specialized or domain-specific vocabulary
   - Lexical Density: Ratio of content words (nouns, verbs, adjectives, adverbs) to function words (pronouns, prepositions, articles, etc.)

2. Syntactic Complexity (*Weight: 20%*): Analyze sentence-structure complexity using these metrics
   - Sentence Length: Average number of words per sentence
   - Clause Density: Number of clauses per sentence
   - Subordination: Frequency and depth of subordinate clauses
   - Passive Voice: Proportion of sentences in passive voice
   - Syntactic Variety: Diversity of sentence structures
   - Embedding Depth: How deeply clauses are nested within one another

3. Conceptual Density (*Weight: 25%*): Assess the density and abstraction level of ideas presented
   - Concept Count: Number of distinct concepts, ideas, or arguments introduced
   - Concept Abstraction: Level of concreteness vs. abstraction of concepts
   - Conceptual Networks: Complexity of relationships between concepts
   - Information Density: Amount of information conveyed per paragraph
   - Theoretical Complexity: Depth of theoretical constructs presented

4. Domain Specificity (*Weight: 15%*): Evaluate how much specialized domain knowledge is required
   - Background Knowledge: Prerequisite knowledge assumed by the text
   - Domain Vocabulary: Concentration of field-specific terminology
   - Conceptual Familiarity: How familiar concepts would be to general readers
   - Specialized References: References to domain-specific methods, theories, or figures
   - Audience Specificity: How targeted the text is to specialists vs. general readers

5. Logical Structure (*Weight: 10%*): Analyze the complexity of reasoning patterns
   - Argument Structure: Complexity of argumentative or explanatory structure
   - Logical Operations: Presence of conditional, causal, comparative reasoning
   - Inference Requirements: Extent to which the reader must infer rather than being told explicitly
   - Logical Connections: Clarity and complexity of connections between ideas
   - Reasoning Chains: Length and complexity of logical chains

6. Contextual Dependencies (*Weight: 10%*): Assess how much the text relies on external context
   - Intertextual References: References to other texts or knowledge sources
   - Cultural Knowledge: Required cultural or historical background
   - Implicit Information: Amount of information that remains unstated yet necessary
   - Presuppositions: Assumptions the text makes about reader knowledge
   - Discourse Context: Degree to which meaning depends on broader discourse context

**Text to Evaluate**
`{text}`
**Required Output Format**
Only return a JSON object with the following structure:

```
{
  "lexical_complexity": {
    "score": <0-10 number>
  },
  "syntactic_complexity": {
    "score": <0-10 number>
  },
  "conceptual_density": {
    "score": <0-10 number>
  },
  "domain_specificity": {
    "score": <0-10 number>
  },
  "logical_structure": {
    "score": <0-10 number>
  },
  "contextual_dependencies": {
    "score": <0-10 number>
  },
  "final_weighted_score":
  <calculated final score as decimal>,
  "normalized_complexity_score":
  <rounded to one decimal place, e.g. 4.5>
}
```

## J  THE USE OF LARGE LANGUAGE MODELS (LLMs)

We utilized GPT-4.1-mini API to generate complexity scores for training contexts. Each 1024-token context was evaluated across six semantic dimensions: lexical complexity, syntactic complexity, conceptual density, domain specificity, logical structure, and contextual dependencies, producing normalized complexity scores from 0 to 10. Additionally, we employed LLMs for language polishing and grammatical refinement of the manuscript.

