# OpenReview forum: "AdaptiveK Sparse Autoencoders: Dynamic Sparsity Allocation for Interpretable LLM Representations"
_ICLR.cc/2026/Conference — ICLR 2026 Conference Withdrawn Submission_

### Official Review · Reviewer_Nrdm · 2025-10-15

**Soundness:** 2
**Presentation:** 3
**Contribution:** 4
**Rating:** 6
**Confidence:** 4

**Summary:**

This paper makes use of the notion of complexity of LLM textual inputs to estimate the ideal dimensionality of an SAE.
Probes trained to predict complexity from LLM activations work with good correlation to complexity and decent errors.
The paper introduces a 3 phase training routine for AdaptiveK SAEs, and runs evaluations on many LLMs up to 14b parameters.
They also propose a case study, and show interpretability on it.

**Strengths:**

- impressive results on FVU/L0 and CosSim/L0 and Loss/L0, which improves the pareto frontier by a lot, and moves the paper beyond incremental improvement territory
- clear motivation
- well placed in the literature, as all the methods this paper beats shows

**Weaknesses:**

Methodological:
- gpt4.1 mini is a weak way to score complexity, and you train the probes on gpt4.1 mini's complexity estimates.
- only studies classic SAEs and not other variants like cross coders
- % of interpretable features is omitted (only a case study is provided), and this is an issue because you can't control sparsity (which is tied to interpretability) to help in case your interpretability is low

If the above are addressed I will raise my score to 8.

Style:
- no quantified improvement in abstract and intro
- topk in figure 1 is trivial
- writing is not succinct (but understandable since ICLR expects you to hit 9 pages)

**Questions:**

Have you tried getting some data labellers for complexity?

Have you looked at other methods to assess complexity like: average word length or sentence length or Flesch reading-ease tests?

Have you tried assessing human interpretability on a small sample of 100 features, or using something like eleuther auto interp, or ideally both?

---

### Official Review · Reviewer_Rvgd · 2025-10-19

**Soundness:** 2
**Presentation:** 3
**Contribution:** 2
**Rating:** 2
**Confidence:** 3

**Summary:**

The authors introduce a new type of SAE, AdaptiveK SAE, which adapts the degree of sparsity (the k in TopK) depending on the complexity of the prompt being processed. They use a clever approach to determine the prompt complexity, leveraging the fact that complexity is represented linearly in LLM activations. While the idea is clever and fills a significant gap in the literature, I am concerned about the quality of their evaluations, and feel generally uncertain about whether the claims in the paper are well-supported by the evidence presented. I hope the authors address these concerns, because I do like the idea and I hope that after rebuttals I’ll be able to increase my score.

**Strengths:**

1. The paper’s core idea (using the model’s own representation of complexity to guide the degree of sparsity) is remarkably clever. It fills a major gap in the literature using a promising approach. The conceptual contribution here is quite large.
2. The evaluation of whether LLMs represent complexity linearly seems very well-executed, I appreciate the completeness of e.g. section 3.1.2.
3. The paper does a wide range of analysis, including evaluations with SAEBench, comparisons to many baselines, and testing on many different models.
4. The paper is generally well-written and easy to follow (with only a small handful of exceptions which I highlight in the Weaknesses section).

**Weaknesses:**

Major concerns (Addressing these would make me increase my score. The idea of the paper is good, but the evaluations need to improve)

1. The authors claim that one of the major benefits of their approach is that it removes the necessity of expensive hyperparameter sweeps to find the right value of k. However, in equation 7 they introduce at least 2 new hyperparameters, k\_min and k\_max. (c\_min and c\_max may also be intended as hyperparameters, though I imagine they could be set automatically by picking the empirical min and max across a few batches; the authors do not clarify this.) So they remove the need for tuning 1 hyperparam, but add the need for tuning 2 hyperparams, which significantly weakens the usefulness. It appears that in the experiments, they simply pick arbitrary numbers for these hyperparameters (line 292), which they do not justify.
2. The baselines they use for reconstruction fidelity are suspiciously weak. E.g. in Figure 6(a), the authors seem to be claiming that standard ReLU SAEs with L0 \~= 100 only explain about 50% of the variance in Pythia-70M, which seems implausible given the rest of the literature on this topic. (Also in Figure 6(a), the unexplained variance of gated SAEs goes *up* as their L0 goes up, which seems very strange.)
3. I’m very confused by Figure 8 and related claims around line 455\. Are the authors simply selecting one feature in an AdaptiveK SAE, one feature from a TopK SAE, observing that the former is more semantically complex, and claiming that this suggests AdaptiveK SAEs are in general more capable of finding semantically interesting features? This would be extremely cherry-picked, a sample size of 1 does not tell us anything meaningful when it comes to SAE features. The authors should clarify what the claim here is.
4. The choice to “train on the last token representation of each context” (line 353\) seems concerning from a training efficiency perspective. It suggests that the authors do a forward pass on 1,000+ tokens but then only use one of them to train, compared to other SAE architectures which would train on all of them. Since forward passes are a substantial part of the computational expenses of training SAEs, it seems to me that this choice would significantly increase the cost of training AdaptiveK SAE. (In addition, the authors appear to train on a very small number of tokens (250,000) which likely means their SAEs are quite undertrained; this makes me a bit more sceptical of the evaluations.)

Moderate concerns (addressing these may make me increase my score)

5. When assessing reconstruction fidelity, I strongly recommend including metrics which measure downstream performance when the reconstructed activations are substituted into the LLM (e.g. a CE loss score, or a CE delta).
6. When scoring context complexity, where do the “evaluation dimensions” and their weights come from? Why is “Lexical Complexity” twice as important as “Contextual Dependencies”?
7. The authors only use one LLM (GPT-4.1-mini) to assess complexity, and they use this as the ground truth. While using LLM scoring is a reasonable approach, the authors should explore the extent to which different LLMs give different scores. Would e.g. Claude give significantly different scores here? It would be good to include some score correlations between different LLMs.
8. The authors should ideally include something like an autointerp score ([https://arxiv.org/pdf/2410.13928](https://arxiv.org/pdf/2410.13928)), or something similar to convince us that this architecture still has features which are for the most part interpretable.
9. What is the justification for the third phase of training? This seems quite strange conceptually, it makes me slightly worried that there could be difficult-to-diagnose training pathologies which are muddying the waters in the evaluations. To address this, the authors could include an ablation where they train SAEs without this third stage and report the results in an appendix.
10. They take the hidden activation only at the last token. Is that justified? It is true that the last sequence position is the only one that can “see” the entire prompt, but it could also be the case that it doesn’t represent the earlier parts of the prompt with sufficient richness, in effect over-weighting the later parts of the prompt. One could run an experiment where you construct prompts where the first half is very complicated while the second half is simple, and vice versa and see what the probe predicts; if it differs significantly between these two setups that would indicate that the probe’s position in the sequence is indeed causing issues.

Mild concerns (these are unlikely, in and of themselves, to make me increase my score, but they could help in combination with other things)

11. Why are they including an L1 sparsity penalty in the SAE’s loss function? Shouldn’t the TopK already take care of sparsity on its own?
12. Why have they decided to include an auxiliary loss function? It would be nice to have a brief justification (and ideally an appendix where they do some experiments without this, as an ablation; though this is quite low-priority).
13. On lines 266–269, they claim that AdaptiveK helps with feature suppression, can they clarify how it does this over standard TopK SAEs (or alternatively cut that claim)?

**Questions:**

See the Weaknesses section. Along with addressing the concerns, I would appreciate more information on my bullet points 3, 4, 6, 9, and 11, as these points in the paper feel particularly confusing.

**Details Of Ethics Concerns:**

No concerns

---

### Official Review · Reviewer_2NPK · 2025-10-26

**Soundness:** 2
**Presentation:** 1
**Contribution:** 2
**Rating:** 2
**Confidence:** 4

**Summary:**

The authors show that LLMs linearly encode semantic complexity of tokens. They then use this signal to determine what the k should be for each token for a TopK SAE where the k is set in that way. They show that this approach outperforms previous methods using a Pareto curve plot (at least for one point on the curve) which is quite a nice result. This gives some (admittedly weak) evidence for their hypothesis that semantic complexity and representational capacity required are highly related and also may tentatively represent an improvement to the SAE training methodology.
Overall the authors have a nice idea in terms of complexity which they execute in order to present a new SAE methodology and their approach is quite theoretically compelling even if they're empirical results could use more clarity.

The paper's presentation leaves some to be desired and there are some comments and questions below that if addressed would make this a stronger paper and I may be willing to increase my score.

Note: I got the error message "Unknown error, contact the admin" when I tried to access and download the code. Could the authors please check the link provided?

**Strengths:**

- Figure 2 is well presented
- On the Pareto plots their approach does indeed seem to outperform the previous approaches. It would be nice to see a Pareto frontier from their approach - e.g. as you adjust the average sparsity how does their approach vary compared to other approaches?
- The approach that the authors look at is intuitively interesting
- If the authors claim that less hyperparameter tuning is required for their approach is true then this is a great quality of this approach

**Weaknesses:**

- Clarity of written exposition is quite low with several sections which were not well explained. Section 2.1 in particular would be hard to follow for someone not familiar with the literature
- Figure 1 is difficult to interpret is the error fixed for all of the SAEs presented?
    - If so then their approach is outperformed by others.
    - If not then having fewer activated features is not a virtue if it could be that the error of their approach is higher. Here we really need a Pareto plot rather than this bar plot for the idea to be informative
- Figure 3 is fairly hard to read - I'd suggest having a single chart and putting the others in the appendix for readability
- Figure 4 is way too small and the legend is unreadable at standard size. Since this doesn't seem to add much I'd suggest removing this figure
- Algorithm 1 is not very detailed and so doesn't provide much more useful information than the text - I'd recommend either adding detail to the algorithm or removing the algorithm altogether
- For Figure 5 all charts should have at least 3 labelled positions on the y axis otherwise it is not possible for readers to ascertain by how much the proposed approach is preferred over previous methods
- If the input latents x are normalised then L2 loss and Cosine similarity are a linear transformation away from each other and so it is not generally useful to plot both metrics here
- The citation for VocabProj is not correct - it points to a survey paper
- It's not clear what the advantage of Figure 8 and 9 is. What we would like to see here is a quantitive analysis that shows that your method outperforms previous methods in terms of human interpretability but it appears that instead we just have a small number of cherry picked examples which are not convincing in showing that your method is an improvement
- It would be useful to show ablations on their method, e.g. is the third step required? How sensitive is the method to hyperparameters etc
- Table 2 is not properly formatted
- I agree with the overall intuition that sparsity rates should vary by complexity. This seems important.
    - However, it seems that BatchTopK (Bussman et al 2024) and Feature Choice SAEs (Ayonrinde 2024) both do this though. Yes they have a fixed K across the batch but if the batch is made up of different tokens which have different levels of complexity then we should generally expect this to give some allocation via sparsity
    - Hence this paper isn't the first to have this variable top-k by complexity, the contribution should be that there's a better method of leveraging complexity for adaptivity than prior work
    - Also note that Ayonrinde, 2024 (Adaptive Sparse Allocation with Mutual Choice & Feature Choice Sparse Autoencoders) is probably the most similar work to this and I don't believe that it's cited here
- I don't believe RQ4 is answered in this work. RQ1 is somewhat cursorily answered - we would like to know if the k value found here is actually the optimal in some sense which doesn't seem to be done here (though they have shown that the k values chosen are better than previous works which is a pretty good result)

**Questions:**

- Do the authors do any analysis to ensure that the complexity scores provided by the LLM is actually representing semantic complexity as we would understand this?
    - In particular the evidence for the probe working is that it aligns with the LLM scores but this is not the "true complexity score" as claimed it is another model!
- From Table 1 the authors claim that the Linear probe performs comparatively which is true but it still does perform somewhat worse than the MLP - it is debatable how much of a success this is for the Linear Representation Hypothesis - what do the authors think about this? (E.g. we could imagine the MLP overfitting and hence performing worse than the linear model for truly linear data)
- Nowhere in the paper does it explain what the auxiliary loss is - what is this?
- RQ2 and RQ3 appear to be the same? What is the difference?
- Are you aware of the paper "Interpretability as Compression" which introduces MDL-SAEs and the description length objective for SAEs? This might be a nice theoretical and evaluation setting for you to present your approach within
- Some of the appendix results look intuitively compelling, it might be worth considering moving some of these to the main paper and for the ones where it underperforms giving some details on why you think this is the case

---

### Official Review · Reviewer_h8YX · 2025-10-30

**Soundness:** 2
**Presentation:** 3
**Contribution:** 3
**Rating:** 4
**Confidence:** 4

**Summary:**

This paper introduces AdaptiveK Sparse Autoencoders (AdaptiveK SAE), a framework that dynamically adjusts sparsity levels based on input text complexity. The authors first demonstrate that text complexity is linearly encoded in LLM activations via linear probes, and then use this relationship to adaptively set the number of active features k_{\text{adp}} for each input. Experiments across multiple LLMs (70M–14B) show improved reconstruction fidelity, explained variance, cosine similarity, and interpretability compared to fixed-sparsity SAEs such as TopK and BatchTopK.

**Strengths:**

- Novel direction: relating feature activation quantity to text complexity, potentially refining SAE sparsity allocation.
- The use of a linear probe to estimate complexity offers a simple and scalable mechanism.
- Experiments cover a wide range of LLM scales and provide both reconstruction and interpretability analyses.
- The idea could inform future work on dynamic sparsity or resource allocation in interpretability models.

**Weaknesses:**

The paper’s core idea is conceptually meaningful and experimentally supported, but:
- The connection between k_{\text{adp}} and the complexity score (Eq. 7) is heuristic and lacks clear motivation or theoretical justification.
- The explained variance (EV) formula in line 395 is incorrect, and this error appears to propagate into key results (e.g., Pareto frontier in Fig. 6). You can refer to https://arxiv.org/pdf/2410.20526 for the right one.
- Unexpectedly poor baseline performance: In Figure 6b, baseline FVU (1 - EV) reaches approximately 0.4 at an L0 sparsity around 100, whereas GemmaScope reports significantly better results(around 0.2) under comparable conditions. While differences in evaluation frameworks may account for some variation, the magnitude of this discrepancy is substantial. Furthermore, the erroneous calculation of EV in this work would systematically inflate FVU values, thereby exacerbating concerns about the reliability of the comparative analysis.
- The evaluation focuses only on context-level (sentence-level) complexity, which may limit generalization and obscure token-level interpretability.
- Some analyses (e.g., Pareto frontiers) appear overly optimistic given the issues above.
- Adaptive k still contains a lot of hyperparameters, and finding the right hyperparameters still requires a lot of effort.
Minor
- Duplicated phrase: “logical structure” (lines 160–161).
- Use “Appendix H” instead of “Section H” (line 351).

**Questions:**

1. Can you provide a clearer motivation or derivation for Eq. (7)? Why use a sigmoid mapping, and how were parameters s, k_{\min}, k_{\max} chosen?
2. Please correct the EV computation and improve Figs. 6–7. How do results change under the correct definition?
3. Were all baseline methods trained with the same evaluation protocol and data subset? Some discrepancies suggest otherwise.
4. How does the model perform if the complexity score is noisy (e.g., random perturbations)? Is the approach robust to estimation errors?

---

### Note · Authors · 2026-01-05

I have read and agree with the venue's withdrawal policy on behalf of myself and my co-authors.